# Time-Consistent Investment and Consumption Strategies under a General Discount Function

Ishak Alia [1], Farid Chighoub [1] , Nabil Khelfallah [1] and Josep Vives [2,*]

1   Laboratory of Applied Mathematics, University Mohamed Khider, P.O. Box 145, Biskra 07000, Algeria;
    ishak.alia@hotmail.com (I.A.); f.chighoub@univ-biskra.dz or chighoub_farid@yahoo.fr (F.C.);
    nabilkhelfallah@yahoo.fr (N.K.)
2   Departament de Matemàtiques i Informàtica, Universitat de Barcelona, Gran Via 585, 08007 Barcelona, Spain
*   Correspondence: josep.vives@ub.edu

**Abstract:** In the present paper, we investigate the Merton portfolio management problem in the context of non-exponential discounting, a context that gives rise to time-inconsistency of the decision-maker. We consider equilibrium policies within the class of open-loop controls that are characterized, in our context, by means of a variational method which leads to a stochastic system that consists of a flow of forward-backward stochastic differential equations and an equilibrium condition. An explicit representation of the equilibrium policies is provided for the special cases of power, logarithmic and exponential utility functions.

**Keywords:** stochastic optimization; investment-consumption problem; Merton portfolio problem; non-exponential discounting; time inconsistency; equilibrium strategies; stochastic maximum principle

**MSC:** 93E20; 60H30; 93E99; 60H10

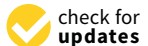



## 1. Introduction

In recent years, there has been a renewed attention in time inconsistency for optimality problems, as well as financial models. Generally, time inconsistency arises in several optimality problems when the optimal strategy selected at some time $s$ is no longer optimal at time $t > s$. In other words, a strategy is time-inconsistent when the decision-maker at future time $t > s$ is tempted to deviate from the strategy determined at time $s$. It is well known that an optimization problem gives raise to time-inconsistent strategies when the dynamic programming principle cannot be applied, and Bellman's principle does not hold.

In practice, an important illustration of time-inconsistent problem is the mean-variance selection problem, where the time inconsistency is due to the fact that there is a nonlinear function of the expectation of the final wealth in the objective criterion. Another important problem which produces a time-inconsistent behavior is the investment-consumption problem with non-exponential discounting. This was the case studied by Strotz (1955–1956), where the time inconsistency arises by the fact that the initial point in time enters in a crucial manner the objective criterion.

The common assumption in classical investment-consumption problems under discounted utility is that the discount rate is assumed to be constant over time which leads to the discount function be exponential. This assumption provides the possibility to compare outcomes occurring at different times by discounting future utility by some constant factor. But, on the other hand, results from experimental studies contradict this assumption, indicating that discount rates for the near future are much lower than discount rates for the time further away in future. Ainslie (1995) established experimental studies on human and animal behavior and found that discount functions are almost hyperbolic; that is, they decrease like a negative power of time rather than an exponential. Loewenstein and Prelec (1992) showed that economic decision-makers are impatient about choices in the short term but

are more patient when choosing between long-term alternatives; therefore, a hyperbolic type discount function would be more realistic.

Unfortunately, as soon as a discount function is non-exponential, discounted utility models become time-inconsistent in the sense that they do not admit the Bellman's optimality principle. Consequently, the classical dynamic programming approach may not be applicable to solve these problems. According to Strotz (1955–1956), there are two basic ways of handling time inconsistency in non-exponential discounted utility models. In the first one, under the notion of naive agents, every decision is taken without taking into account that their preferences will change in the near future. The agent at time $t \in [0, T]$ will solve the problem as a standard optimal control problem with initial condition $X(t) = x_t$. If we suppose that the naive agent at time 0 solves the problem, his or her solution corresponds to the so-called pre-commitment solution, in the sense that it is optimal as long as the agent can pre-commit his or her future behavior at time $t = 0$.

Kydland and Prescott (1997) indeed argue that a pre-committed strategy may be economically meaningful in certain circumstances. The second approach consists in the formulation of a time-inconsistent decision problem as a non-cooperative game between incarnations of the decision-maker at different instants of time. Nash equilibrium of these strategies are then considered to define the new concept of solution to the original problem. Strotz (1955–1956) was the first who proposed a game theoretic formulation to handle the dynamic time inconsistent optimal decision problem on the deterministic Ramsey problem; see Ramsey (1928). Then, by capturing the idea of non-commitment, and letting the commitment period being infinitesimally small, he provided a primitive notion of Nash equilibrium strategy.

## 1.1. Related Works

Further work along this line in continuous and discrete time has been done by Pollak (1968); Phelps and Pollak (1968); Goldman (1980); Barro (1990); and Krusell and Smith (2003). Keeping the same game theoretic approach, Ekeland and Lazrak (2008) and Marín-Solano and Navas (2010) treated the optimal consumption problem where the utility involves a non-exponential discount function in the deterministic framework. They characterized the equilibrium strategies by a value function that has to satisfy a certain "extended Hamilton–Jacobi–Bellman (HJB) equation", which is a non-linear differential equation displaying a non-local term that depends on the global behavior of the solution. In this situation, every decision at time $t$ is taken by a $t-$agent which represents the incarnation of the controller at time $t$ and is referred in Marín-Solano and Navas (2010) as a "sophisticated $t-$agent".

Björk and Murgoci (2010) extended the idea to the stochastic setting where the controlled dynamic is driven by a quite general class of Markov process and a fairly general objective function. Yong in Yong (2011), by a discretization of time, studied a class of time inconsistent deterministic linear quadratic models and derive equilibrium controls via some class of Riccati-Voltera equations. Yong (2012), also by a discretization of time, investigated a general discounting time inconsistent stochastic optimal control problem and characterizes a feedback time-consistent Nash equilibrium control via the so-called "equilibrium HJB equation".

The Nash equilibrium solution to the mean-variance problem was established first by Basak and Chabakauri (2010) and then extended to a further general class of time-inconsistent problems by Björk and Murgoci (2010). Other papers on the consistent planning approach for the mean-variance problem are Hu et al. (2012); Czichowsky (2013); and Björk et al. (2014).

Concerning equilibrium strategies for an optimal consumption-investment problem with a general discount function, Ekeland and Pirvu (2008) were the first to investigate Nash equilibrium strategies where the price process of the risky asset is driven by geometric Brownian motion. They characterized the equilibrium strategies through the solutions of a flow of BSDEs, and they show, for an special form of the discount function, that

the BSDEs reduce to a system of two ODEs, which has a solution. Ekeland et al. (1995) added life insurance to the investor's portfolio and they characterize the equilibrium strategy by an integral equation. In Yong (2012), the case of time-inconsistent consumption-investment problem under a power utility function is discussed. Following Yong's approach, Zhao et al. (2014) studied the consumption-investment problem with a general discount function and a logarithmic utility function. Recently, Zou et al. (2014) investigated equilibrium consumption-investment decisions for Merton's portfolio problem with stochastic hyperbolic discounting.

As for the comparison between different methods to time inconsistency, Wang and Forsyth (2012) evaluate time-consistent against pre-commitment strategies and compare their related efficient frontiers for a mean-variance optimization problem. The evaluation among the naive and the sophisticated approaches is given by Chen et al. (2014), who study the optimal dividend model of an insurance company in the existence of time inconsistency created by non-exponential discount factor. Cong and Oosterlee (2016) found a relation linking the time-consistent and the pre-commitment investment strategies in a defined contribution pension scheme. Cui et al. (2017) emphasize the shortcomings of pre-commitment and game theoretical strategies, and examine a self-coordination strategy that aims at corresponding global concern and local interests of the decision-maker. Van Staden et al. (2018) consider the pre-commitment and the time-consistent policies in the attendance of realistic investment constraints. Bensoussan et al. (2019) evaluate the produce of constraints on the value function of both pre-commitment and game theoretical approaches, and discover the unexpected result that for the game theoretical approach the occurrence of constraints can improve the payoff, whereas, for the pre-commitment approach, this paradox does not arise. Menoncin and Vigna (2020) evaluate the contribution pension scheme and prove that the dynamically optimal policy reacts better to extreme scenarios of market returns.

Time-inconsistent consumption-investment problem with a non-exponential discount function and a general utility function. We use the game theoretic approach to handle the time inconsistency in the same perspective as Björk and Murgoci (2010). Noting that, the game perspective that we will consider is as follows: We first consider a game with one player at each point *t* in time. This player represents the incarnation of the decision maker at time *t* and can be referred to as "player *t*". This *t* − *th* player can control the system only at time *t* by taking his/her strategies. A control process is then viewed as a complete description of the chosen strategies of all players in the game. The reward to player *t* is given by utility function of an investment-consumption optimization problem. From this description, we introduce the concept of a "perfect Nash equilibrium strategy" of the game. This is an admissible control process satisfying some admissibility conditions.

We focus on a variational technique approach leading to a version of a necessary and sufficient condition for equilibrium, which involves a flow of forward-backward stochastic differential equations (FBSDEs) along with a certain equilibrium condition. We also present a verification theorem that covers some possible examples of utility functions. Then, by decoupling the flow of the FBSDEs, we derive a closed-loop representation of the equilibrium strategies via a parabolic non-linear partial differential equation (PDE). We show that within a special form of the utility function (logarithmic, power, and exponential) the PDE reduces to a system of ODEs which has an explicit solution.

### 1.2. Novelty and Contribution

Different from Marín-Solano and Navas (2010) and Ekeland and Pirvu (2008), where the authors derived explicit solutions for special forms of the discount factor, in our model, the non-exponential discount function is in a fairly general form. Moreover, we consider equilibrium strategies in the open-loop sense, as defined in Hu et al. (2012) and Hu et al. (2015), which is different from most of the existing literature on this topic. Note also that the time-inconsistency, in our paper, arises from a non-exponential discounting in the objective function, while the works Hu et al. (2012) and Hu et al. (2015) are concerned

with a quite different kind of time-inconsistency which is caused by the presence of non-linear terms of expectations in the terminal cost. On other hand, the objective functional, in our paper, is not reduced to the quadratic form as in Hu et al. (2012) and Hu et al. (2015).

We accentuate that, different from most of the existing literature on this topic, where some feedback equilibrium strategies are derived via several very complicated highly non-linear integro-differential equations, an explicit representation of the equilibrium strategies are obtained in our work via simple ODEs. In addition, this method can provide the necessary and sufficient conditions to characterize the equilibrium strategies, while the extended HJB techniques can create, in general, only the sufficient condition in the form of a verification theorem that characterizes the equilibrium strategies.

### 1.3. Structure of the Paper

The rest of the paper is organized as follows. In Section 2, we formulate the problem and give the necessary notations and preliminaries. In Section 3, we present the main results of the paper, Theorem 1 and Theorem 2, that characterize the equilibrium decisions by some necessary and sufficient conditions. In Section 4, we derive an explicit representation of the equilibrium consumption-investment strategy. Section 5 is devoted to some comparisons with existing results in the literature. The paper ends with an Appendix containing some proofs.

## 2. Problem Formulation

In what follows, we assume that $W(\cdot) = (W_1(\cdot), \ldots, W_d(\cdot))$ is a d-dimensional standard Brownian motion defined on a filtered probability space $(\Omega, \mathcal{F}, \mathbb{F}, \mathbb{P})$, such that $\mathbb{F} := (\mathcal{F}_t)_{t \in [0,T]}$ is a natural filtration that satisfies the usual conditions, in particular, $\mathcal{F}_0$ contains all $\mathbb{P}$-null sets and $\mathcal{F}_T = \mathcal{F}$ for an arbitrarily fixed finite time horizon $T > 0$. Recall that $\mathcal{F}_t$ stands for the information available up to time $t$, and any decision made at time $t$ is based on this information.

### 2.1. Notations

Throughout this paper, we use the following notations: $M^\top$: the transpose of the vector (or matrix) $M$, $\langle \chi, \zeta \rangle$ : the inner product of $\chi$ and $\zeta$, that is, $\langle \chi, \zeta \rangle := tr(\chi^T \zeta)$. For a function $f$, we denote by $f_x$ (resp. $f_{xx}$) the first (resp. the second) derivative of $f$ with respect to the variable $x$.

For any Euclidean space $E$ with Frobenius norm $|\cdot|$, we let, for any $t \in [0, T]$,

- $\mathbb{L}^p(\Omega, \mathcal{F}_t, \mathbb{P}; E)$ : for any $p \geq 1$, the set of $E-$valued $\mathcal{F}_t-$measurable random variables $X$, such that $\mathbb{E}\big[|X|^p\big] < \infty$.
- $\mathcal{L}_{\mathcal{F}}^2(t, T; E)$ : the space of $E-$valued, $(\mathcal{F}_s)_{s \in [t,T]}-$adapted continuous processes $Y(\cdot)$, with

$$\|Y(\cdot)\|_{\mathcal{L}_{\mathcal{F}}^2(t,T;E)} = \sqrt{\mathbb{E}\left[\sup_{s \in [t,T]} |Y(s)|^2\right]} < \infty.$$

- $\mathcal{M}_{\mathcal{F}}^p(t, T; E)$ : for any $p \geq 1$, the space of $E-$valued, $(\mathcal{F}_s)_{s \in [t,T]}-$adapted processes $Z(\cdot)$, with

$$\|Z(\cdot)\|_{\mathcal{M}_{\mathcal{F}}^p(t,T;E)} = \mathbb{E}\left[\int_0^T |Z(s)|^p ds\right]^{\frac{1}{p}} < \infty.$$

### 2.2. Financial Market

Consider an individual facing the inter-temporal consumption and portfolio problem where the market environment consists of one riskless and $d$ risky securities. The risky securities are stocks and their prices are modeled as Itô processes. Namely, for $i = 1, 2, .., d$, the price $S_i(s)$, for $s \in [0, T]$, of the i-th risky asset, satisfies

$$dS_i(s) = S_i(s)\left(\mu_i(s)ds + \sum_{j=1}^{d} \sigma_{ij}(s)dW_j(s)\right), \tag{1}$$

with $S_i(0) > 0$, for $i = 1, 2, ..., d$, and the coefficients $\mu_i(\cdot)$ and $\sigma_i(\cdot) = (\sigma_{i1}(\cdot), \ldots, \sigma_{id}(\cdot))$, for $i = 1, .., d$, are $\mathbb{F}$−progressively measurable processes with values in $\mathbb{R}$ and $\mathbb{R}^d$, respectively. For brevity, we use $\mu(\cdot) = (\mu_1(\cdot), \mu_2(\cdot), \ldots, \mu_d(\cdot))$ to denote the drift rate vector and $\sigma(\cdot) = (\sigma_{ij}(\cdot))_{1 \le i,j \le d}$ to denote the random volatility matrix.

The riskless asset, or the savings account, has the price process $S_0(s)$, for $s \in [0, T]$, governed by

$$dS_0(s) = r_0(s)S_0(s)ds, \ S_0(0) = 1, \tag{2}$$

where $r_0(\cdot)$ is a process with values in $[0, \infty)$ that represents the interest rate. We assume that $\mathbb{E}[\mu_i(t)] > r_0(t) \ge 0$, $dt - a.e.$, for $i = 1, 2, .., d$. This is a very natural assumption since, otherwise, nobody is willing to invest in the risky stocks.

### 2.3. Investment-Consumption Policies and Wealth Process

Starting from an initial capital $x_0 > 0$ at time 0, during the time horizon $[0, T]$, the decision-maker is allowed to dynamically invest in the stocks, as well as in the bond and consuming. A consumption-investment strategy is described by a $(d + 1)$-dimensional stochastic process $u(\cdot) = (c(\cdot), u_1(\cdot), \ldots, u_d(\cdot))^\top$, where $c(s)$ represents the consumption rate at time $s \in [0, T]$ and $u_i(s)$, for $i = 1, 2, .., d$, represents the amount invested in the *i-th* risky stock at time $s \in [0, T]$. The process $u_I(\cdot) = (u_1(\cdot), \ldots, u_d(\cdot))^\top$ is called an investment strategy. The amount invested in the bond at time $s$ is

$$X^{x_0, u}(s) - \sum_{i=1}^{d} u_i(s),$$

where $X^{x_0, u}(\cdot)$ is the wealth process associated with the strategy $u(\cdot)$ and the initial capital $x_0$. The evolution of $X^{x_0, u}(\cdot)$ can be described as

$$\begin{cases} dX^{x_0, u}(s) = \left(X^{x_0, u}(s) - \sum_{i=1}^{d} u_i(s)\right)\dfrac{dS_0(s)}{S_0(s)} + \sum_{i=1}^{d} u_i(s)\dfrac{dS_i(s)}{S_i(s)} - c(s)ds, \text{ for } s \in [0, T], \\ X^{x_0, u}(0) = x_0. \end{cases}$$

Accordingly, the wealth process solves the SDE:

$$\begin{cases} dX^{x_0, u}(s) = \left\{r_0(s)X^{x_0, u}(s) + u_I(s)^\top r(s) - c(s)\right\}ds \\ \qquad\qquad + u_I(s)^\top \sigma(s)dW(s), \text{ for } s \in [0, T], \\ X^{x_0, u}(0) = x_0. \end{cases} \tag{3}$$

where $r(\cdot) = (\mu_1(\cdot) - r_0(\cdot), \ldots, \mu_d(\cdot) - r_0(\cdot))^\top$.

As time evolves, it is natural to consider the controlled stochastic differential equation parametrized by $(t, \xi) \in [0, T] \times \mathbb{L}^2(\Omega, \mathcal{F}_t, \mathbb{P}; \mathbb{R})$ and satisfied by $X(\cdot) = X^{t, \xi}(\cdot; u(\cdot))$,

$$\begin{cases} dX(s) = \left\{r_0(s)X(s) + u_I(s)^\top r(s) - c(s)\right\}ds + u_I(s)^\top \sigma(s)dW(s), \text{ for } s \in [t, T], \\ X(t) = \xi. \end{cases} \tag{4}$$

**Definition 1** (Admissible Strategy). *A strategy $u(\cdot) = \left(c(\cdot), u_I(\cdot)^\top\right)^\top$ is said to be admissible over $[t, T]$ if $u(\cdot) \in \mathcal{M}_{\mathcal{F}}^1(t, T; \mathbb{R}) \times \mathcal{M}_{\mathcal{F}}^2\left(t, T; \mathbb{R}^d\right)$ and for any $(t, \xi) \in [0, T] \times \mathbb{L}^2(\Omega, \mathcal{F}_t, \mathbb{P}; \mathbb{R})$, the equation (4) has a unique solution $X(\cdot) = X^{t, \xi}(\cdot; u(\cdot))$.*

We impose the following assumption about the coefficients.

**(H1)** Processes $r_0(\cdot)$, $r(\cdot)$ and $\sigma(\cdot)$ are uniformly bounded and moreover we assume the following uniform ellipticity condition:

$$\sigma(s)\sigma(s)^{\top} \geq \epsilon I_d, \ ds - a.e, \ d\mathbb{P} - a.s.$$

for some $\epsilon > 0$, where $I_d$ denotes the identity matrix on $\mathbb{R}^{d \times d}$.

Under **(H1)**, for any $(t, \xi, u(\cdot)) \in [0, T] \times \mathbb{L}^2(\Omega, \mathcal{F}_t, \mathbb{P}; \mathbb{R}) \times \mathcal{M}^1_{\mathcal{F}}(t, T; \mathbb{R}) \times \mathcal{M}^2_{\mathcal{F}}(t, T; \mathbb{R}^d)$, the state equation (4) has a unique solution $X(\cdot) \in \mathcal{L}^2_{\mathcal{F}}(t, T; \mathbb{R})$. Moreover, we have the estimate

$$\mathbb{E}\left[\sup_{t \leq s \leq T} |X(s)|^2\right] \leq K\left(1 + \mathbb{E}\left[|\xi|^2\right]\right), \tag{5}$$

for some positive constant $K$. In particular, for $t = 0$, $x_0 > 0$ and $u(\cdot) = \left(c(\cdot), u_I(\cdot)^{\top}\right)^{\top} \in \mathcal{M}^1_{\mathcal{F}}(0, T; \mathbb{R}) \times \mathcal{M}^2_{\mathcal{F}}(0, T; \mathbb{R}^d)$, the state equation (3) has a unique solution $X^{x_0, u}(\cdot) \in \mathcal{L}^2_{\mathcal{F}}(0, T; \mathbb{R})$, and the following estimate holds:

$$\mathbb{E}\left[\sup_{0 \leq s \leq T} |X^{x_0, u}(s)|^2\right] \leq K\left(1 + |x_0|^2\right). \tag{6}$$

### *2.4. General Discounted Utility Function*

Most of financial-economics works have considered that the rate of time preference is constant (exponential discounting). However, there is growing evidence to suggest that this may not be the case. In this section, we discuss the general discounting preferences. We also introduce the basic modeling framework of Merton's consumption and portfolio problem. We refer the reader to Ainslie et al. (1991), Karatzas et al. (1987), Merton (1969), Merton (1971), and Pliska (1986) for more detail about the classical Merton model.

### 2.4.1. Discount Function

As soon as discounting is non-exponential, most papers work with special form of the non-exponential discount factor. Differently from these works, we consider a general form of the discount factor.

**Definition 2.** *A discount function $\lambda(\cdot) : [0, T] \to \mathbb{R}$ is a continuous and deterministic function satisfying $\lambda(0) = 1$, $\lambda(s) > 0$ $ds - a.e.$ and $\int_0^T \lambda(s) ds < \infty$.*

**Remark 1.** *Some examples of discount functions are given in many papers, such as exponential discount functions (see Merton (1969) and Merton (1971)), mixture of exponential functions (see Ekeland and Pirvu (2008)), and hyperbolic discount functions (see Zhao et al. (2014)).*

### 2.4.2. Utility Functions and Objective

In order to evaluate the performance of a consumption-investment strategy, the decision-maker derives utility from inter-temporal consumption and final wealth. Let $\varphi(\cdot)$ be the utility of inter-temporal consumption and $h(\cdot)$ the utility of the terminal wealth at some non-random horizon $T$ (which is a primitive of the model). Then, for any $(t, \xi) \in [0, T] \times \mathbb{L}^2(\Omega, \mathcal{F}_t, \mathbb{P}; \mathbb{R})$, the investment-consumption optimization problem is reduced to maximize the utility function $J(t, \xi, .)$ given by

$$J(t, \xi, u(\cdot)) = \mathbb{E}^t\left[\int_t^T \lambda(s - t)\varphi(c(s))ds + \lambda(T - t)h(X(T))\right], \tag{7}$$

over $u(\cdot) \in \mathcal{M}^1_{\mathcal{F}}(t, T; \mathbb{R}) \times \mathcal{M}^2_{\mathcal{F}}(t, T; \mathbb{R}^d)$, subject to (4), where $\mathbb{E}^t[\cdot] = \mathbb{E}[\cdot | \mathcal{F}_t]$. We restrict ourselves to utility functions which satisfy the following conditions.

**(H2)** The maps $\varphi(\cdot), h(\cdot) : \mathbb{R} \to \mathbb{R}$ are strictly increasing, strictly concave and satisfy the integrability condition

$$\mathbb{E}\left[\int_0^T |\varphi(c(s))| ds + |h(X(T))|\right] < \infty.$$

Noting that, most literature on the necessary and/or sufficient optimality conditions in stochastic control problem that considers more strong conditions about the coefficients, in which the derivatives $\varphi_{xx}(\cdot)$ and $h_{xx}(\cdot)$ are bounded or have linear or quadratic growth; see, e.g., Yong and Zhou (1999). It is also worth mentioning that , several papers impose an $L_p$ bounds on the control process for $p > 2$. Those restrictions make it impossible to apply the stochastic maximum principle approach directly to study the consumption–investment problem. Certainly, in that important problem, the optimal control is not necessarily $L_p$-integrable, and the derivatives of the running cost and the terminal cost do not necessarily follow the global polynomial growth conditions.

In this paper, we overcome the technical difficulties mentioned above and treat some limiting procedures. To do so, let us introduce further technical integrability conditions of the utilities, which will be used in the proof of the main result.

**(H3)** The maps $\varphi(\cdot), h(\cdot)$ are twice continuously differentiable functions, so all the derivatives $\varphi_x(\cdot), h_x(\cdot), \varphi_{xx}(\cdot)$ and $h_{xx}(\cdot)$ are continuous.

**(H4)** For all admissible strategy pairs, there exists a constant $p > 1$ such that

$$\mathbb{E}\left[\int_0^T |\varphi_x(c(s))|^p ds + |h_x(X(T))|^p\right] < \infty,$$

$$\mathbb{E}\left[\int_0^T \sup_{\eta \in \mathbb{R}, |\eta| \leq M} |\varphi_{xx}(c(s) + \eta)|^p ds\right] < \infty, \text{ for } M \geq 0.$$

If we write $W^\star(s) = \left(0, W(s)^\top\right)^\top$, and we denote $B(s) = \left(-1, r(s)^\top\right)^\top$, $\Gamma = \left(1, 0_{\mathbb{R}^d}^\top\right)^\top$ and

$$D(s) = \begin{pmatrix} 0 & 0_{\mathbb{R}^d}^\top \\ 0_{\mathbb{R}^d} & \sigma(s) \end{pmatrix},$$

then the optimal control problem associated with (4) and (7) is equivalent to maximize

$$J(t, \xi, u(\cdot)) = \mathbb{E}^t\left[\int_t^T \lambda(s - t)\varphi\left(\Gamma^\top u(\cdot)\right) ds + \lambda(T - t)h(X(T))\right], \tag{8}$$

subject to

$$\begin{cases} dX(s) = \left\{r_0(s)X(s) + u(s)^\top B(s)\right\} ds + u(s)^\top D(s) dW^\star(s), \text{ for } s \in [t, T], \\ X(t) = \xi. \end{cases} \tag{9}$$

over $u(\cdot) \in \mathcal{M}^1_{\mathcal{F}}(t, T; \mathbb{R}) \times \mathcal{M}^2_{\mathcal{F}}(t, T; \mathbb{R}^d)$.

2.4.3. Time Inconsistency

Let us first note that the optimal policies, although they exist, will not be time-consistent in general. First of all, as an illustration, let us consider the model in (8)–(9) with logarithmic utility functions. We suppose that the financial market consists of one riskless asset and $d$ risky assets. Arguing as in Ekeland and Pirvu (2008), we can prove

that, if the agent is naive and starts with a given positive wealth $x$, at some instant $t$, then, by the standard dynamic programming approach, the value function associated with this stochastic control problem solves the following Hamilton–Jacobi–Bellman equation.

$$
\begin{cases}
V_s^t(s,x) + \sup_{(c,u_I) \in \mathbb{R}^{d+1}} \left\{ \left( r_0(s)X(s) + u_I^\top r(s) - c \right) V_x^t(s,x) + \frac{1}{2} u_I^\top \sigma(s)\sigma(s)^\top u_I V_{xx}^t(s,x) \right. \\
\qquad \left. + \frac{\lambda'(s-t)}{\lambda(s-t)} V^t(s,x) + \varphi(c) \right\} = 0, \text{ for } s \in [t,T], \\
V^t(T,x) = h(x).
\end{cases}
\tag{10}
$$

The HJB equation contains the term $\dfrac{\lambda'(s-t)}{\lambda(s-t)}$, which depends not only on the current time $s$ but also on initial time $t$, so the optimal policy will depend on $t$, as well. Indeed, the first order necessary conditions yield the $t-$optimal policy

$$
\bar{u}_I^t(s) = r(s) \left( \sigma(s)\sigma(s)^\top \right)^{-1} \frac{V_x^t(s,x)}{V_{xx}^t(s,x)},
$$
$$
\bar{c}^t(s) = \varphi^{-1} \left( V_x^t(s,x) \right).
$$

Let us consider the following example: $\varphi(x) = h(x) = \log x$. The naive agent for the initial pair $(0, x_0)$ solves the problem, assuming that the discount rate of time preference will be $\lambda(s)$, for $s \in [0,T]$, and the optimal consumption strategy will be

$$
\bar{c}^{0,x_0}(s) = \left[ 1 + \int_s^T \exp\left\{ \lambda(r-s) + \log\left( \frac{\lambda(r)}{\lambda(s)} \right) \right\} dr \right]^{-1}, \text{ for } s \in [0,T].
$$

This solution corresponds to the so-called pre-commitment solution, in the sense that it is optimal as long as the agent can pre-commit (by signing a contract, for example) his or her future behavior at time $t = 0$. If there is no commitment, the 0-agent will take the action $\bar{c}^{0,x_0}(s)$, but, in the near future, the $\epsilon$-agent will change his decision rule (time-inconsistency) to the solution of the HJB equation (10) with $t = \epsilon$. In this case, the optimal control trajectory for $s > \epsilon$ will be changed to $\bar{c}^{\epsilon,x_\epsilon}(s)$, given by

$$
\bar{c}^{\epsilon,x_\epsilon}(s) = \bar{c}^{\epsilon,\bar{X}(\epsilon)}(s) = \left[ 1 + \int_s^T \exp\left\{ \lambda(r-s) + \log\left( \frac{\lambda(r-\epsilon)}{\lambda(s-\epsilon)} \right) \right\} dr \right]^{-1}, \text{ for } s \in [\epsilon,T].
$$

If $\lambda(t) = e^{-\delta t}$, where $\delta > 0$ is the constant discount rate, then

$$
\bar{c}^{0,x_0}_{|[\epsilon,T]}(s) = \bar{c}^{\epsilon,x_\epsilon}(s), \text{ for } s \in [\epsilon,T];
$$

hence, the optimal consumption plan is time consistent. As soon as discount function is non-exponential

$$
\bar{c}^{0,x_0}_{|[\epsilon,T]}(s) \neq \bar{c}^{\epsilon,x_\epsilon}(s), \text{ for } s \in [\epsilon,T].
$$

Then, the optimal consumption plan is not time consistent. In general, the solution for the naive agent will be constructed by solving the family of HJB equations (10) for $t \in [0,T]$, and patching together the "optimal" solutions $\bar{c}^{t,x_t}(t)$. If the agent is sophisticated, things become more complicated. The standard HJB equation cannot be used to construct the solution, and a new method is required in what follows.

## 3. Equilibrium Strategies

It is well known that the problem described above by (8)–(9) turns out to be time inconsistent in the sense that it does not satisfy the Bellman optimality principle, since a restriction of an optimal control for a specific initial pair on a later time interval might

not be optimal for that corresponding initial pair. For a more detailed discussion see Ekeland and Pirvu (2008) and Yong (2012). Due to the lack of time consistency, we consider open-loop Nash equilibrium controls instead of optimal controls. As in Hu et al. (2012), we first consider an equilibrium by local spike variation, given, for $t \in [0, T]$, an admissible consumption-investment strategy $\hat{u}(\cdot) \in \mathcal{M}^1_{\mathcal{F}}(t, T; \mathbb{R}) \times \mathcal{M}^2_{\mathcal{F}}\left(t, T; \mathbb{R}^d\right)$. For any $\mathbb{R}^{d+1}-$valued, $\mathcal{F}_t-$measurable and bounded random variable $v$ and for any $\varepsilon > 0$, define

$$u^\varepsilon(s) := \begin{cases} \hat{u}(s) + v, & \text{for } s \in [t, t+\varepsilon), \\ \hat{u}(s), & \text{for } s \in [t+\varepsilon, T]. \end{cases} \tag{11}$$

We have the following definition.

**Definition 3** (Open-loop Nash equilibrium). *An admissible strategy $\hat{u}(\cdot) \in \mathcal{M}^1_{\mathcal{F}}(t, T; \mathbb{R}) \times \mathcal{M}^2_{\mathcal{F}}\left(t, T; \mathbb{R}^d\right)$ is an open-loop Nash equilibrium strategy if*

$$\liminf_{\varepsilon \downarrow 0} \frac{1}{\varepsilon} \left\{ J\left(t, \hat{X}(t), u^\varepsilon(\cdot)\right) - J\left(t, \hat{X}(t), \hat{u}(\cdot)\right) \right\} \leq 0, \tag{12}$$

*for any $t \in [0, T]$, where $\hat{X}$ is the equilibrium wealth process that solves the SDE*

$$\begin{cases} d\hat{X}(s) = \left\{ r_0(s)\hat{X}(s) + \hat{u}(s)^\top B(s) \right\} ds + \hat{u}(s)^\top D(s)dW^\star(s), \text{ for } s \in [t, T], \\ \hat{X}(t) = \xi. \end{cases} \tag{13}$$

*3.1. A Necessary and Sufficient Condition for Equilibrium Controls*

　　In this paper, we follow an alternative approach, which is essentially a necessary and sufficient condition for equilibrium. In the same spirit of proving the stochastic Pontryagin's maximum principle for equilibrium in Hu et al. (2012) for the case of linear quadratic models, we derive this condition by a second-order expansion in the spike variation.

　　Now, we introduce the adjoint equations involved in the characterization of open-loop Nash equilibrium controls.

3.1.1. Adjoint Processes

　　Let $\hat{u}(\cdot) = \left( \hat{c}(\cdot), \hat{u}_I(\cdot)^\top \right)^\top \in \mathcal{M}^1_{\mathcal{F}}(0, T; \mathbb{R}) \times \mathcal{M}^2_{\mathcal{F}}\left(0, T; \mathbb{R}^d\right)$ an admissible strategy and denote by $\hat{X}(\cdot) \in \mathcal{L}^2_{\mathcal{F}}(0, T; \mathbb{R})$ the corresponding wealth process. For each $t \in [0, T]$, we introduce the first order adjoint equation defined on the time interval $[t, T]$, and satisfied by the pair of processes $(p(\cdot; t), q(\cdot; t))$ as follows

$$\begin{cases} dp(s; t) = -r_0(s)p(s; t)ds + q(s; t)^\top dW(s), \text{ for } s \in [t, T], \\ p(T; t) = \lambda(T-t)h_x\left(\hat{X}(T)\right), \end{cases} \tag{14}$$

where $q(\cdot; t) = (q_1(\cdot; t), \ldots, q_d(\cdot; t))^\top$. Under the assumption **(H1)**, the BSDE (14) is uniquely solvable in $\mathcal{L}^2_{\mathcal{F}}(t, T; \mathbb{R}) \times \mathcal{M}^2_{\mathcal{F}}\left(t, T; \mathbb{R}^d\right)$. Moreover, there exists a constant $K > 0$ such that, for any $t \in [0, T]$, we have the following estimate

$$\|p(\cdot; t)\|^2_{\mathcal{L}^2_{\mathcal{F}}(t, T; \mathbb{R})} + \|q(\cdot; t)\|^2_{\mathcal{M}^2(t, T; \mathbb{R}^d)} \leq K\left(1 + \xi^2\right). \tag{15}$$

　　The second order adjoint equation is defined on the time interval $[t, T]$ and satisfied by the pair of processes $(P(\cdot; t), Q(\cdot; t)) \in \mathcal{L}^2_{\mathcal{F}}(t, T; \mathbb{R}) \times \mathcal{M}^2_{\mathcal{F}}\left(t, T; \mathbb{R}^d\right)$ as follows:

$$\begin{cases} dP(s; t) = -2r_0(s)P(s; t)ds + Q(s; t)^\top dW(s), \text{ for } s \in [t, T], \\ P(T; t) = \lambda(T-t)h_{xx}\left(\hat{X}(T)\right), \end{cases} \tag{16}$$

where $Q(\cdot;t) = (Q_1(\cdot;t),\ldots,Q_d(\cdot;t))^\top$. Under **(H1)**, the above BSDE has a unique solution $(P(\cdot;t),Q(\cdot;t)) \in \mathcal{L}^2_{\mathcal{F}}(t,T;\mathbb{R}) \times \mathcal{M}^2_{\mathcal{F}}(t,T;\mathbb{R}^d)$. Moreover, we have the following representation for $P(\cdot;t)$:

$$P(s;t) = \mathbb{E}^s\left[\lambda(T-t)e^{\int_s^T 2r_0(\tau)d\tau}h_{xx}\big(\hat{X}(T)\big)\right], \text{ for } s \in [t,T]. \tag{17}$$

Indeed, if we define the function $\Theta(\cdot,t)$, for each $t \in [0,T]$, as the fundamental solution of the linear ODE

$$\begin{cases} d\Theta(\tau,t) = r_0(\tau)\Theta(\tau,t)d\tau, \text{ for } \tau \in [t,T], \\ \Theta(t,t) = 1, \end{cases} \tag{18}$$

and we apply the Itô's formula to $\tau \to P(\tau;t)\Theta(\tau,t)^2$ on $[t,T]$, by taking conditional expectations, we obtain (17). Note that, since $h_{xx}\big(\hat{X}(T)\big) \leq 0$, then $P(s;t) \leq 0, ds - a.e.$

### 3.1.2. A Characterization of Equilibrium Strategies

The following theorem is the first main result of this work, and it provides a necessary and sufficient condition for equilibrium. As we have said before, the proof is inspired by Hu et al. (2012) and Hu et al. (2015).

First, we define the process $\tilde{q}(s;t) = \left(0,q(s;t)^\top\right)^\top$, and we introduce the following notations:

$$\mathcal{H}(s;t) \triangleq p(s;t)B(s) + D(s)\tilde{q}(s;t) + \lambda(s-t)\varphi_x\left(\Gamma^\top\hat{u}(s)\right)\Gamma, \tag{19}$$

and, for a certain $\theta \in [0,1]$,

$$\mathcal{A}^\varepsilon(s;t) \triangleq \begin{pmatrix} \lambda(s-t)\varphi_{xx}\left(\Gamma^\top\left(\hat{u}(s) + \theta v 1_{[t,t+\varepsilon)}(s)\right)\right) & 0_{\mathbb{R}^d}^\top \\ 0_{\mathbb{R}^d} & \sigma(s)\sigma(s)^\top P(s;t) \end{pmatrix}. \tag{20}$$

**Theorem 1.** *Let **(H1)–(H4)** hold. Given an admissible strategy $\hat{u}(\cdot) \in \mathcal{M}^1_{\mathcal{F}}(0,T;\mathbb{R}) \times \mathcal{M}^2_{\mathcal{F}}(0,T;\mathbb{R}^d)$, let for any $t \in [0,T]$, the process*

$$(p(\cdot;t),q(\cdot;t)) \in \mathcal{L}^2_{\mathcal{F}}(t,T;\mathbb{R}) \times \mathcal{M}^2_{\mathcal{F}}(t,T;\mathbb{R}^d)$$

*be the unique solution to the BSDE (14). Then, $\hat{u}(\cdot)$ is an equilibrium consumption-investment strategy, if and only if, the following condition holds*

$$\mathcal{H}(t;t) = 0, d\mathbb{P}-a.s., dt - a.e. \tag{21}$$

In order to derive the proof of this theorem, let us, first of all, derive some technical results. First, denote by $\hat{X}^\varepsilon(\cdot)$ the solution of the state equation corresponding to $u^\varepsilon(\cdot)$. Since the coefficients of the controlled state equation are linear, using the standard perturbation approach (see, e.g., Yong and Zhou (1999)), we have

$$\hat{X}^\varepsilon(s) - \hat{X}(s) = y^{\varepsilon,v}(s) + z^{\varepsilon,v}(s), \text{ for } s \in [t,T], \tag{22}$$

where, for any $\mathbb{R}^{d+1}-$valued, $\mathcal{F}_t-$measurable, and bounded random variable $v$, and for any $\varepsilon \in [0,T-t)$, $y^{\varepsilon,v}(\cdot)$ and $z^{\varepsilon,v}(\cdot)$, solve, respectively, the following linear stochastic differential equations:

$$\begin{cases} dy^{\varepsilon,v}(s) = r_0(s)y^{\varepsilon,v}(s)ds + v^\top D(s)1_{[t,t+\varepsilon)}(s)dW^\star(s), \text{ for } s \in [t,T], \\ y^{\varepsilon,v}(t) = 0, \end{cases} \tag{23}$$

and

$$\begin{cases} dz^{\varepsilon,v}(s) = \left\{ r_0(s)z^{\varepsilon,v}(s) + v^\top B(s)1_{[t,t+\varepsilon)}(s) \right\}ds, \text{ for } s \in [t,T], \\ z^{\varepsilon,v}(t) = 0. \end{cases} \tag{24}$$

**Proposition 1.** *Let **(H1)–(H4)** hold. For any $t \in [0,T]$, the following estimates hold for any $k \geq 1$:*

$$\mathbb{E}^t \left[ \sup_{s \in [t,T]} |y^{\varepsilon,v}(s)|^{2k} \right] = O\left(\varepsilon^k\right), \tag{25}$$

$$\mathbb{E}^t \left[ \sup_{s \in [t,T]} |z^{\varepsilon,v}(s)|^{2k} \right] = O\left(\varepsilon^{2k}\right), \tag{26}$$

$$\mathbb{E}^t \left[ \sup_{s \in [t,T]} |y^{\varepsilon,v}(s) + z^{\varepsilon,v}(s)|^{2k} \right] = O\left(\varepsilon^k\right). \tag{27}$$

*In addition, we have the following equality:*

$$\begin{aligned} J\left(t, \hat{X}(t), u^\varepsilon(\cdot)\right) &- J\left(t, \hat{X}(t), \hat{u}(\cdot)\right) \\ &= \int_t^{t+\varepsilon} \mathbb{E}^t \left[ \langle \mathcal{H}(s;t), v \rangle + \frac{1}{2}\langle \mathcal{A}^\varepsilon(s;t)v, v \rangle \right] ds + o(\varepsilon). \end{aligned} \tag{28}$$

**Proof.** See Appendix A. □

Now, we present the following technical lemma needed later. The proof follows an argument adapted from Hamaguchi (2019).

**Lemma 1.** *Under assumptions **(H1)–(H4)**, there exists a sequence $\left(\varepsilon_n^t\right)_{n\in\mathbb{N}} \subset (0, T-t)$ satisfying $\varepsilon_n^t \to 0$ as $n \to \infty$, such that*

*(1)* $\lim_{n\to\infty} \dfrac{1}{\varepsilon_n^t} \int_t^{t+\varepsilon_n^t} \mathbb{E}^t[\mathcal{H}(s;t)]ds = \mathcal{H}(t;t),\ d\mathbb{P} - a.s,\ dt - a.e.$

*(2)* $\lim_{n\to\infty} \dfrac{1}{\varepsilon_n^t} \int_t^{t+\varepsilon_n^t} \mathbb{E}^t \left[ \mathcal{A}^{\varepsilon_n^t}(s;t) \right] ds = \mathcal{A}^0(t;t),\ d\mathbb{P} - a.s,\ dt - a.e.$

**Proof.** See Appendix A. □

**Proof of Theorem 1.** Given an admissible strategy

$$\hat{u}(\cdot) \in \mathcal{M}_{\mathcal{F}}^1(0,T;\mathbb{R}) \times \mathcal{M}_{\mathcal{F}}^2\left(0,T;\mathbb{R}^d\right),$$

for which (21) holds, according to Lemma 1, we have from (28) that, for any $t \in [0,T]$, and for any $\mathbb{R}^{d+1}$−valued, $\mathcal{F}_t$−measurable, and bounded random variable $v$, there exists a sequence $\left(\varepsilon_n^t\right)_{n\in\mathbb{N}} \subset (0, T-t)$ satisfying $\varepsilon_n^t \to 0$ as $n \to \infty$, such that

$$\begin{aligned} \lim_{n\to 0} \frac{1}{\varepsilon_n^t} \left\{ J\left(t, \hat{X}(t), u^\varepsilon(\cdot)\right) - J\left(t, \hat{X}(t), \hat{u}(\cdot)\right) \right\} &= \langle \mathcal{H}(t;t), v \rangle + \frac{1}{2}\left\langle \mathcal{A}^0(t;t)v, v \right\rangle, \\ &= \frac{1}{2}\left\langle \mathcal{A}^0(t;t)v, v \right\rangle, \\ &\leq 0, \end{aligned}$$

where we have used in the last inequality the fact that, under the concavity condition of $\varphi(\cdot)$ and $h(\cdot)$, it follows $\left\langle \mathcal{A}^0(t;t)v, v \right\rangle \leq 0$. Hence, $\hat{u}(\cdot)$ is an equilibrium strategy.

Conversely, assume that $\hat{u}(\cdot)$ is an equilibrium strategy. Then, by (12), together with (28) and Lemma 1, for any $(t, u) \in [0, T] \times \mathbb{R}^{d+1}$, the following inequality holds:

$$\langle \mathcal{H}(t; t), u \rangle + \frac{1}{2} \Big\langle \mathcal{A}^0(t; t) u, u \Big\rangle \leq 0. \tag{29}$$

Now, we define $\forall (t, u) \in [0, T] \times \mathbb{R}^{d+1}$,

$$\Phi(t, u) = \langle \mathcal{H}(t; t), u \rangle + \frac{1}{2} \Big\langle \mathcal{A}^0(t; t) u, u \Big\rangle.$$

Clearly, $\Phi(\cdot, \cdot)$ is well defined. In fact, it is a second order polynomial in terms of the components of vector $u$. Easy manipulations show that the inequality (29) is equivalent to

$$\Phi(t, 0) = \max_{u \in \mathbb{R}^{d+1}} \Phi(t, u), \; d\mathbb{P} - a.s, \forall t \in [0, T]. \tag{30}$$

So, it is easy to see that the maximum condition (30) leads to the following condition: $\forall t \in [0, T]$,

$$\Phi_u(t, 0) = \mathcal{H}(t; t) = 0, \; d\mathbb{P} - a.s. \tag{31}$$

According to Lemma 1, the expression (21) follows immediately. This completes the proof. $\square$

### 3.2. A Characterization of Equilibrium Strategies by Verification Argument

In classical (time-consistent) stochastic control theory, the sufficient condition of optimality is of significant importance for computing optimal controls. It says that, if an admissible control satisfies the maximum condition of the Hamiltonian function, then the control is indeed optimal for the stochastic control problem. This allows one to solve examples of optimal control problems, where one can find a smooth solution to the associated adjoint equation.

The aim of the following theorem is to characterize the open-loop equilibrium pair only by a sufficient condition of equilibrium. Let us introduce an alternative to **(H3)** hypothesis:

**(H3')** The maps $\varphi(\cdot), h(\cdot)$ are continuously differentiable, and the first order derivatives $\varphi_x(\cdot), h_x(\cdot)$ are continuous.

Then, we have the following theorem:

**Theorem 2.** *Let (H1), (H2) and (H3') hold. Given an admissible strategy $\hat{u}(\cdot) \in \mathcal{M}_{\mathcal{F}}^1(0, T; \mathbb{R}) \times \mathcal{M}_{\mathcal{F}}^2\left(0, T; \mathbb{R}^d\right)$, let, for any $t \in [0, T]$, the process*

$$(p(\cdot; t), q(\cdot; t)) \in \mathcal{L}_{\mathcal{F}}^2(t, T; \mathbb{R}) \times \mathcal{M}_{\mathcal{F}}^2\left(t, T; \mathbb{R}^d\right)$$

*be the unique solution to the BSDE (14). Then, $\hat{u}(\cdot)$ is an equilibrium consumption-investment strategy, if the following condition holds:*

$$\mathcal{H}(t; t) = 0, \; d\mathbb{P}-a.s., \; dt - a.e. \tag{32}$$

**Proof.** Suppose that $\hat{u}(\cdot)$ is an admissible control for which the condition (32) holds. In addition, for any $t \in [0, T]$ and $\varepsilon \in [0, T - t)$, we consider $u^\varepsilon(\cdot)$ by (11). Then, we have the following difference.

$$J\big(t, \hat{X}(t), \hat{u}(\cdot)\big) - J\big(t, \hat{X}(t), u^\varepsilon(\cdot)\big)$$
$$= \mathbb{E}^t \left[ \int_t^T \lambda(s - t) \Big( \varphi\big(\Gamma^\top \hat{u}(s)\big) - \varphi\big(\Gamma^\top u^\varepsilon(s)\big) \Big) ds + \lambda(T - t) \big( h(\hat{X}(T)) - h(\hat{X}^\varepsilon(T)) \big) \right].$$

Noting that, by the concavity of $h(\cdot)$, we have

$$\mathbb{E}^t\big[\lambda(T-t)\big(h(\hat{X}(T))-h(\hat{X}^\varepsilon(T))\big)\big] \geq \mathbb{E}^t\Big[\lambda(T-t)\big(\hat{X}(T)-\hat{X}^\varepsilon(T)\big)^T h_x(\hat{X}(T))\Big].$$

Accordingly, by the terminal condition in the BSDE (14), we obtain that

$$J\big(t,\hat{X}(t),\hat{u}(\cdot)\big) - J\big(t,\hat{X}(t),u^\varepsilon(\cdot)\big)$$
$$\geq \mathbb{E}^t\bigg[\int_t^T \lambda(s-t)\Big(\varphi\big(\Gamma^\top \hat{u}(s)\big) - \varphi\big(\Gamma^\top u^\varepsilon(s)\big)\Big)ds + \big(\hat{X}(T)-\hat{X}^\varepsilon(T)\big)^T p(T;t)\bigg]. \quad (33)$$

By applying Ito's formula to $s \mapsto \big(\hat{X}(s)-\hat{X}^\varepsilon(s)\big)^T p(s;t)$ on $[t,T]$, we get

$$\mathbb{E}^t\Big[\big(\hat{X}(T)-\hat{X}^\varepsilon(T)\big)^T p(T;t)\Big] = \mathbb{E}^t\bigg[\int_t^T (\hat{u}(s)-u^\varepsilon(s))^T (B(s)p(s;t)+D(s)\tilde{q}(s;t))ds\bigg]. \quad (34)$$

By the concavity of $\varphi(\cdot)$, we find

$$\mathbb{E}^t\bigg[\int_t^T \lambda(s-t)\Big(\varphi\big(\Gamma^\top \hat{u}(s)\big) - \varphi\big(\Gamma^\top u^\varepsilon(s)\big)\Big)ds\bigg]$$
$$\geq \mathbb{E}^t\bigg[\int_t^T \lambda(s-t)\Big\langle \varphi_x\big(\Gamma^\top \hat{u}(s)\big)\Gamma, \hat{u}(s)-u^\varepsilon(s)\Big\rangle ds\bigg]. \quad (35)$$

By taking (34) and (35) in (33), it follows that

$$J\big(t,\hat{X}(t),u^\varepsilon(\cdot)\big) - J\big(t,\hat{X}(t),\hat{u}(\cdot)\big)$$
$$\leq \mathbb{E}^t\bigg[\int_t^T \Big\langle B(s)p(s;t)+D(s)\tilde{q}(s;t)+\lambda(s-t)\varphi_x\big(\Gamma^\top \hat{u}(s)\big)\Gamma, u^\varepsilon(s)-\hat{u}(s)\Big\rangle ds\bigg]$$
$$= \mathbb{E}^t\bigg[\int_t^{t+\varepsilon} \langle \mathcal{H}(s;t),v\rangle ds\bigg].$$

Now, dividing both sides by $\varepsilon$ and taking the limit when $\varepsilon$ vanishes, by Lemma 1, we conclude that $\hat{u}(\cdot)$ is an equilibrium control. $\square$

**Remark 2.** *The purpose of the sufficient condition of optimality is to find an optimal control by computing the difference $J(\hat{u}(\cdot)) - J(u(\cdot))$ in terms of the Hamiltonian function, where $u(\cdot)$ is an arbitrary admissible control. Here, the spike variation perturbation* (11) *plays a key role in deriving the sufficient condition for equilibrium strategies, which reduces to the computation of the difference $J\big(t,\hat{X}(t),\hat{u}(\cdot)\big) - J\big(t,\hat{X}(t),u^\varepsilon(\cdot)\big)$, without the necessity to achieving the second order expansion in the spike variation.*

## 4. Equilibrium When the Coefficients Are Deterministic

Theorems 1 and 2 show that one can obtain equilibrium consumption-investment strategies by solving a system of FBSDEs which is not standard since the "flow" of the unknown process $(p(\cdot;t),q(\cdot;t))_{t\in[0,T]}$ is involved. Moreover, there is an additional constraint that act on the "diagonal" (i.e., when $s = t$) of the flow. As far as we know, the explicit solvability of this type of equations remains an open problem, except for some particular form of the utility function. However, we are able to solve quite thoroughly this problem when the parameters $r_0(\cdot)$, $\mu(\cdot)$ and $\sigma(\cdot)$ are deterministic functions. In this section, we define what we mean by an equilibrium rule, and then we derive a parabolic backward PDE. Our PDE is comparable with the one obtained in Marín-Solano and Navas (2010) and Ekeland and Pirvu (2008), for some particular discount functions in a finite horizon with different utility functions.

In this section, let us look at the Merton's portfolio problem with general discounting and deterministic parameters. At first, we consider the following parabolic backward partial differential equation:

$$
\begin{cases}
\theta_t(t,x) + \theta_x(t,x)\left(r_0(t)x - r(t)^\top\Sigma(t)r(t)\dfrac{\theta(t,x)}{\theta_x(t,x)} - \mathcal{I}(\lambda(T-t)\theta(t,x))\right) \\
+ \dfrac{1}{2}\theta_{xx}(t,x)r(t)^\top\Sigma(t)r(t)\left(\dfrac{\theta(t,x)}{\theta_x(t,x)}\right)^2 + \theta(t,x)r_0(t) = 0, \ (t,x) \in [0,T]\times\mathbb{R}, \\
\theta(T,x) = h_x(x),
\end{cases}
\tag{36}
$$

where we denote by $\mathcal{I}(\cdot)$ the inverse function of the strictly decreasing marginal derivative utility $\varphi_x(\cdot)$ and $\Sigma(s) \equiv \left(\sigma(s)\sigma(s)^\top\right)^{-1}$.

We have the following verification theorem.

**Theorem 3.** *Let **(H1)–(H4)** hold. If there exists a classical solution*

$$\theta(\cdot,\cdot) \in \mathcal{C}^{1,2}((0,T)\times\mathbb{R},\mathbb{R}) \cap \mathcal{C}([0,T]\times\mathbb{R},\mathbb{R})$$

*of the PDE* (36) *such that the stochastic differential equation,*

$$
\begin{cases}
d\hat{X}(s) = \left\{r_0(s)\hat{X}(s) - r(s)^\top\Sigma(s)r(s)\dfrac{\theta\big(s,\hat{X}(s)\big)}{\theta_x\big(s,\hat{X}(s)\big)} - \mathcal{I}\big(\lambda(T-s)\theta\big(s,\hat{X}(s)\big)\big)\right\}ds \\
\qquad\quad - \dfrac{\theta\big(s,\hat{X}(s)\big)}{\theta_x\big(s,\hat{X}(s)\big)}r(s)^\top\Sigma(s)\sigma(s)dW(s), \ s\in[0,T], \\
\hat{X}(0) = x_0,
\end{cases}
\tag{37}
$$

*has a unique solution $\hat{X}(\cdot)$, in which the following estimate holds*

$$\mathbb{E}\left[\sup_{0\le t\le T}|X(t)|^2\right] \le K\big(1+|x_0|^2\big),$$

*then, the equilibrium consumption-investment strategy $\hat{u}(\cdot) = \left(\hat{c}(\cdot),\hat{u}_I(\cdot)^\top\right)^\top$ is given by*

$$\hat{c}(t) = \mathcal{I}\big(\lambda(T-t)\theta\big(t,\hat{X}(t)\big)\big), \ dt-a.e., \tag{38}$$

$$\hat{u}_I(t) = -\Sigma(t)r(t)\dfrac{\theta\big(t,\hat{X}(t)\big)}{\theta_x\big(t,\hat{X}(t)\big)}, \ dt-a.e. \tag{39}$$

**Proof.** Suppose that $\hat{u}(\cdot) = \left(\hat{c}(\cdot),\hat{u}_I(\cdot)^\top\right)^\top$ is an equilibrium control and denote by $\hat{X}(\cdot)$ the corresponding wealth process. Then, in view of Theorem 1, there exists an adapted process, $\left(\hat{X}(\cdot),(p(\cdot;t),q(\cdot;t))_{t\in[0,T]}\right)$, solution of the following flow of forward-backward SDEs, parametrized by $t \in [0,T]$ :

$$
\begin{cases}
dX(s) = \left\{r_0(s)\hat{X}(s) + \hat{u}_I(s)^\top r(s) - \hat{c}(s)\right\}ds + \hat{u}_I(s)^\top\sigma(s)dW(s), \ s\in[t,T], \\
dp(s;t) = -r_0(s)p(s;t)ds + q(s,t)^\top dW(s), \ 0\le t\le s\le T, \\
\hat{X}(0) = x_0, \ p(T;t) = \lambda(T-t)h_x\big(\hat{X}(T)\big), \ t\in[0,T],
\end{cases}
\tag{40}
$$

with conditions

$$-p(t;t) + \varphi_x(\hat{c}(t)) = 0, \ dt-a.e., \tag{41}$$

$$p(t;t)r(t) + \sigma(t)q(t;t) = 0, \ dt-a.e. \tag{42}$$

From the terminal condition in the first order adjoint process, we consider the following Ansatz

$$p(s;t) = \lambda(T-t)\mathcal{V}(s, \hat{X}(s)), \ \forall \, 0 \le t \le s \le T, \tag{43}$$

for some deterministic function $\mathcal{V}(\cdot, \cdot) \in \mathcal{C}^{1,2}([0, T] \times \mathbb{R}, \mathbb{R})$ such that $\mathcal{V}(T, \cdot) = h_x(\cdot)$.

Applying Itô's formula to (43), it yields

$$dp(s;t) = \lambda(T-t)\left\{\mathcal{V}_s\big(s, \hat{X}(s)\big) + \mathcal{V}_x\big(s, \hat{X}(s)\big)\left(\hat{X}(s)r_0(s) + \hat{u}_I(s)^\top r(s) - \hat{c}(s)\right)\right.$$
$$\left. + \frac{1}{2}\mathcal{V}_{xx}\big(s, \hat{X}(s)\big)\hat{u}_I(s)^\top \sigma(s)\sigma(s)^\top \hat{u}_I(s)\right\}ds$$
$$+ \lambda(T-t)\mathcal{V}_x\big(s, \hat{X}(s)\big)\hat{u}_I(s)^\top \sigma(s)dW(s). \tag{44}$$

Next, comparing the *ds* term in (44) by the ones in the second equation in (40), we deduce that

$$\mathcal{V}_s\big(s, \hat{X}(s)\big) + \mathcal{V}_x\big(s, \hat{X}(s)\big)\left(\hat{X}(s)r_0(s) + \hat{u}_I(s)^\top r(s) - \hat{c}(s)\right)$$
$$+ \frac{1}{2}\mathcal{V}_{xx}\big(s, \hat{X}(s)\big)\hat{u}_I(s)^\top \sigma(s)\sigma(s)^\top \hat{u}_I(s) = -r_0(s)\mathcal{V}\big(s, \hat{X}(s)\big), \tag{45}$$

and by comparing the $dW(s)$ terms we also get

$$q(s,t) = \lambda(T-t)\mathcal{V}_x\big(s, \hat{X}(s)\big)\sigma(s)^\top \hat{u}_I(s). \tag{46}$$

We put the above expressions of $p(s;t)$ and $q(s;t)$ at $s = t$ into (41) and (42), and then

$$\lambda(T-t)\mathcal{V}\big(t, \hat{X}(t)\big) - \varphi_x(\hat{c}(t)) = 0, \tag{47}$$

and

$$\mathcal{V}_x\big(t, \hat{X}(t)\big)\sigma(t)\sigma(t)^\top \hat{u}_I(t) = -r(t)\mathcal{V}\big(t, \hat{X}(t)\big), \tag{48}$$

which leads to the following representation

$$\hat{c}(t) = \mathcal{I}\big(\lambda(T-t)\mathcal{V}\big(t, \hat{X}(t)\big)\big), \ dt - a.e., \tag{49}$$

$$\hat{u}_I(t) = -\Sigma(t)r(t)\frac{\mathcal{V}\big(t, \hat{X}(t)\big)}{\mathcal{V}_x\big(t, \hat{X}(t)\big)}, \ dt - a.e. \tag{50}$$

Then, by taking expressions (49) and (50) into (45), this suggests that $\mathcal{V}(\cdot, \cdot)$ coincides with the solution of the PDE (36), evaluated along the trajectory $\hat{X}(\cdot)$, solution of the state equation. $\square$

**Remark 3.** *Equation* (36) *is comparable with the one in Marín-Solano and Navas (2010) and Ekeland and Pirvu (2008), in which the equilibrium is defined within the class of feedback controls.*

**Remark 4.** *Theorem 3 enables us to derive a suitable equilibrium strategy $\hat{u}_I(t)$, as well as $\hat{c}(t)$, at each $t \in [0, T]$, and this permits us to derive directly an explicit expression of equilibrium control in the cases of power, logarithmic, and exponential utility functions. While the duality approach used in Hamaguchi (2019) permits to characterize a stochastic equilibrium solution in terms of a complicated FBSDE system of a closed form, it does not provide an explicit representation.*

## 5. Special Utility Functions

Equilibrium investment-consumption strategies for Merton's portfolio problem with general discounting and deterministic parameters have been studied in Marín-Solano and Navas (2010); Ekeland and Pirvu (2008); and Yong (2012), among others, in different frameworks. In this section, we discuss some special cases in which the function $\theta(\cdot, \cdot)$ may be separated into functions of time and state variables. Then, one needs only to solve

a system of ODEs in order to completely determine the equilibrium strategies. We will compare our results with some existing ones in the literature.

### 5.1. Power Utility Function

To make the problem (8)–(9) explicitly solvable, we consider power utility functions for the running and terminal costs. That is, $\varphi(c) = \frac{c^\gamma}{\gamma}$ and $h(x) = a\frac{x^\gamma}{\gamma}$, with $a > 0$ and $\gamma \in (0,1)$. In this case, the PDE (36) reduces to

$$
\begin{cases}
\theta_t(t,x) + \theta_x(t,x)\left(r_0(t)x - r(t)^\top\Sigma(t)r(t)\frac{\theta(t,x)}{\theta_x(t,x)} - \frac{\lambda(T-t)^{1-\gamma}}{\theta(t,x)^{\gamma-1}}\right) \\
+ \frac{1}{2}\theta_{xx}(t,x)r(t)^\top\Sigma(t)r(t)\left(\frac{\theta(t,x)}{\theta_x(t,x)}\right)^2 + r_0(t)\theta(t,x) = 0, \ (t,x) \in [0,T] \times \mathbb{R}, \\
\theta(T,x) = ax^{\gamma-1}.
\end{cases}
$$

From the terminal condition, we consider the following trial solution

$$
\theta(s,x) = a\Pi(s)x^{\gamma-1},
$$

for some deterministic function $\Pi(\cdot) \in C^1([0,T],\mathbb{R})$ with the terminal condition $\Pi(T) = 1$. Then, by substituting in (36), we obtain

$$
\begin{cases}
\Pi_t(t) + \left(K(t) + Q(t)\Pi(t)^{\frac{1}{\gamma-1}}\right)\Pi(t) = 0, \text{ for } t \in [0,T], \\
\Pi(T) = 1.
\end{cases}
\tag{51}
$$

where

$$
K(t) \equiv \gamma r_0(t) + \frac{1}{2}\frac{\gamma}{(1-\gamma)}r(t)^\top\Sigma(t)r(t),
\tag{52}
$$

and

$$
Q(t) \equiv (1-\gamma)(a\lambda(T-t))^{\frac{1}{\gamma-1}}.
\tag{53}
$$

It remains to determine the function $\Pi(\cdot)$. First, by the change of variable

$$
\Pi(t) = y(t)^{(1-\gamma)}, \text{ for } t \in [0,T],
\tag{54}
$$

we find that $y(\cdot)$ should solve the following ODE

$$
\begin{cases}
y_t(t) - \frac{K(t)}{(\gamma-1)}y(t) - \frac{Q(t)}{(\gamma-1)} = 0, \text{ for } t \in [0,T], \\
y(T) = 1.
\end{cases}
$$

A variation of constant formula yields to

$$
y(t) = \left(1 - \int_t^T \frac{Q(\tau)}{(\gamma-1)}e^{\int_\tau^T \frac{K(l)}{(\gamma-1)}dl}d\tau\right)\exp\left(-\int_t^T \frac{K(\tau)}{(\gamma-1)}d\tau\right), \text{ for } t \in [0,T],
$$

and, subsequently, we obtain

$$
\Pi(t) = \left(1 - \int_t^T \frac{Q(\tau)}{(\gamma-1)}e^{\int_\tau^T \frac{K(l)}{(\gamma-1)}dl}d\tau\right)^{1-\gamma}\exp\left(\int_t^T K(\tau)d\tau\right), \text{ for } t \in [0,T].
$$

In view of Theorem 3, the representation of the Nash equilibrium strategies (38)–(39) gives

$$\hat{c}(t) = (a\lambda(T-t)\Pi(t))^{\frac{1}{\gamma-1}}\hat{X}(t), \ dt - a.e.,\tag{55}$$

$$\hat{u}_I(t) = \Sigma(t)r(t)\frac{\hat{X}(t)}{(1-\gamma)}, \ dt - a.e.\tag{56}$$

This consumption–investment strategy determines a wealth process given by

$$\begin{aligned}X(t) = \quad &x_0 + \int_0^t \left\{ r_0(s) + \frac{1}{(1-\gamma)}r(s)^\top\Sigma(s)r(s) - (a\lambda(T-s)\Pi(s))^{\frac{1}{\gamma-1}} \right\}\hat{X}(s)ds \\ &+ \int_0^t \frac{\hat{X}(s)}{(1-\gamma)}r(s)^\top\Sigma(s)\sigma(s)dW(s), \ t \in [0,T].\end{aligned}$$

The above solution is comparable with the one obtained by Marín-Solano and Navas (2010); Ekeland and Pirvu (2008); and Yong (2012).

### 5.2. Logarithmic Utility Function

Now, let us analyze the case where $\varphi(c) = \ln(c)$ and $h(x) = a\ln(x)$, with $a > 0$. In this case, the PDE (36) reduces to

$$\begin{cases} \theta_t(t,x) + \theta_x(t,x)\left( r_0(t)x - r(t)^\top\Sigma(t)r(t)\dfrac{\theta(t,x)}{\theta_x(t,x)} - (\lambda(T-t)\theta(t,x))^{-1} \right) \\ +\dfrac{1}{2}\theta_{xx}(t,x)r(t)^\top\Sigma(t)r(t)\left( \dfrac{\theta(t,x)}{\theta_x(t,x)} \right)^2 + r_0(t)\theta(t,x) = 0, \ (t,x) \in [0,T]\times\mathbb{R}, \\ \theta(T,x) = \dfrac{a}{x}. \end{cases}\tag{57}$$

Once again, we know that the solution of (57) will be of the form

$$\theta(t,x) = \Pi(t)\frac{a}{x}, \ \text{for } t \in [0,T],\tag{58}$$

where $\Pi(\cdot) \in C^1([0,T],\mathbb{R})$. By substituting in (57), we get

$$\begin{cases} \Pi_t(t) + \dfrac{1}{a\lambda(T-t)} = 0, \ \text{for } t \in [0,T], \\ \Pi(T) = 1, \end{cases}\tag{59}$$

which is explicitly solved by

$$\Pi(t) = 1 + \int_t^T \frac{1}{a\lambda(T-r)}dr, \ \text{for } t \in [0,T].$$

In view of Theorem 3, the representation of the Nash equilibrium strategies (38)–(39) gives

$$\hat{c}(t) = \left( a\lambda(T-t) + \int_t^T \frac{\lambda(T-t)}{\lambda(T-r)}dr \right)^{-1}\hat{X}(t), \ dt - a.e.,\tag{60}$$

$$\hat{u}_I(t) = \Sigma(t)r(t)\hat{X}(t), \ dt - a.e.\tag{61}$$

This consumption–investment strategy determines a wealth process given by

$$\begin{aligned}X(t) = \quad &x_0 + \int_0^t \left\{ r_0(s) + r(s)^\top\Sigma(s)r(s) - \left( a\lambda(T-s) + \int_s^T \frac{\lambda(T-s)}{\lambda(T-r)}dr \right)^{-1} \right\}\hat{X}(s)ds \\ &+ \int_0^t r(s)^\top\Sigma(s)\sigma(s)\hat{X}(s)dW(s).\end{aligned}$$

### 5.3. Exponential Utility Function

Next, we consider the case where $\varphi(c) = -\dfrac{e^{-\gamma c}}{\gamma}$ and $h(x) = -a\dfrac{e^{-\gamma x}}{\gamma}$, with $a, \gamma > 0$. The terminal condition PDE (36) becomes

$$
\begin{cases}
\theta_t(t,x) + \theta_x(t,x)\left( r_0(t)x - r(t)^\top \Sigma(t) r(t) \dfrac{\theta(t,x)}{\theta_x(t,x)} - \dfrac{1}{\gamma}\ln(\lambda(T-t)\theta(t,x)) \right) \\[2mm]
+ \dfrac{1}{2}\theta_{xx}(t,x) r(t)^\top \Sigma(t) r(t)\left( \dfrac{\theta(t,x)}{\theta_x(t,x)} \right)^2 + r_0(t)\theta(t,x) = 0,\; (t,x) \in [0,T] \times \mathbb{R}, \\[2mm]
\theta(T,x) = ae^{-\gamma x}.
\end{cases}
\tag{62}
$$

We try a solution of the form

$$
\theta(t,x) = ae^{-\gamma(\phi(t)x + \psi(t))}, \text{ for } t \in [0,T],
\tag{63}
$$

where $\phi(\cdot), \psi(\cdot) \in C^1([0,T], \mathbb{R})$ such that $\phi(T) = 1$ and $\psi(T) = 0$. By substituting in (62), we get

$$
\left\{ -\gamma\phi_t(t) + \gamma\phi(t)^2 - \gamma\phi(t)r_0(t) \right\} x - \dfrac{1}{2}r(t)^\top \Sigma(t) r(t)
$$
$$
-\gamma\psi_t(t) - \phi(t)\ln(a\lambda(T-t)) + \gamma\phi(t)\psi(t) + r_0(t) = 0.
$$

This suggests that functions $\phi(\cdot)$ and $\psi(\cdot)$ should solve the following system of equations:

$$
\begin{cases}
\phi_t(t) = -r_0(t)\phi(t) + \phi(t)^2,\; t \in [0,T], \\[2mm]
\psi_t(t) = -\dfrac{1}{\gamma}\phi(t)\ln(a\lambda(T-t)) + \phi(t)\psi(t) - \dfrac{1}{2\gamma}r(t)^\top \Sigma(t) r(t) + \dfrac{1}{\gamma}r_0(t),\; t \in [0,T], \\[2mm]
\phi(T) = 1,\; \psi(T) = 0,
\end{cases}
\tag{64}
$$

which is explicitly solvable for $t \in [0,T]$, by

$$
\phi(t) = \dfrac{e^{\int_t^T r_0(\tau)d\tau}}{1 + \int_t^T e^{\int_l^T r_0(\tau)d\tau}dl},
\tag{65}
$$

and

$$
\psi(t) = e^{-\int_t^T \phi(\tau)d\tau} \int_t^T e^{\int_l^T \phi(\tau)d\tau}\left( \dfrac{1}{\gamma}\phi(l)\ln(\lambda(T-l)a) + \dfrac{1}{2\gamma}r(t)^\top \Sigma(t) r(t) - \dfrac{r_0(l)}{\gamma} \right)dt.
\tag{66}
$$

The representation of the Nash equilibrium strategies (38)–(39) gives

$$
\hat{c}(t) = -\dfrac{1}{\gamma}\ln(a\lambda(T-t)) + \phi(t)\hat{X}(t) + \psi(t),\; dt - a.e.
\tag{67}
$$

$$
\hat{u}_I(t) = \dfrac{1}{\gamma}\Sigma(t) r(t)\phi(t)^{-1},\; dt - a.e.
\tag{68}
$$

This consumption–investment strategy determines a wealth process given by

$$
X(t) = x_0 + \int_0^t \left\{ (r_0(s) - \phi(s))\hat{X}(s) + \dfrac{1}{\gamma}\left( r(s)^\top \Sigma(s) r(s)\phi(s)^{-1} - \ln(a\lambda(T-s)) \right) \right.
$$
$$
\left. -\psi(s) \right\} ds + \int_0^t \dfrac{1}{\gamma}\phi(s)^{-1}r(s)^\top \Sigma(s)\sigma(s)dW(s),\; t \in [0,T].
$$

The above solution is comparable with the ones obtained in Marín-Solano and Navas (2010) by solving an extended Hamilton–Jacobi–Bellman (HJB) equations.

## 6. Special Discount Function

As well documented in Marín-Solano and Navas (2010), an agent making a decision at time $t$ is usually called the $t$-agent, and can act in two different ways: naive and sophisticated. Naive agents make decisions without taking into account that their preferences will change in the near future, and then any $t$-agent will solve the problem as a standard optimal control problem with initial condition $X(t) = x_t$, and his decision will be, in general, time-inconsistent. In order to obtain a time consistent strategy, the $t$-agent should be sophisticated, in the sense of taking into account the preferences of all the $s$-agents, for $s \in [t, T]$. Therefore, the approach to handle the time inconsistency in dynamic decision-making problems is by considering time-inconsistent problems as non-cooperative games with a continuous number of players, in which decisions at every instant of time are selected. The solution to the problem of the agent with non-constant discounting should be constructed by looking for the sub-game perfect equilibrium of the associated game with an infinite number of $t$-agents. In Marín-Solano and Navas (2010), the authors looked for a solution of a sophisticated agent to the modified HJB (which is not a partial differential equation due to the presence of a non-local term). Then, they need to define the Markov equilibrium strategies, while, in our work, and different from Marín-Solano and Navas (2010), we use the open-loop equilibrium strategies. This is a significant difference which leads to obtain an important change in the results.

### 6.1. Exponential Discounting with Constant Discount Rate (Classical Model)

At first, we consider the standard exponential discount function $\lambda(t) = e^{-\delta_0 t}$, $t \in [0, T]$, where $\delta_0 > 0$ is a constant representing the discount rate. In this case, our equilibrium solution for the three cases become:

(1) Logarithmic utility

$$\hat{c}(t) = \frac{1}{ae^{-(T-t)\delta_0} + \int_t^T e^{-(l-t)\delta_0} dl} \hat{X}(t), \; dt - a.e.,$$

$$\hat{u}_I(t) = \Sigma(t) r(t) \hat{X}(t), \; dt - a.e.$$

(2) Power utility

$$\hat{c}(t) = \left( ae^{-(T-t)\delta_0} \right)^{\frac{1}{\gamma-1}} \frac{e^{\int_t^T \frac{K(\tau)}{\gamma - 1} d\tau}}{\left( 1 + \int_t^T \left( ae^{-(T-\tau)\delta_0} \right)^{\frac{1}{\gamma-1}} e^{\int_\tau^T \frac{K(l)}{\gamma - 1} dl} d\tau \right)} \hat{X}(t), \; dt - a.e.,$$

$$\hat{u}_I(t) = \Sigma(t) r(t) \frac{\hat{X}(t)}{(1 - \gamma)}, \; dt - a.e.$$

(3) Exponential utility

$$\hat{c}(t) = -\frac{1}{\gamma} \ln \left( ae^{-(T-t)\delta_0} \right) + \phi(t)\hat{X}(t) + \psi(t), \; dt - a.e.,$$

$$\hat{u}_I(t) = \Sigma(t) r(t) \frac{1}{\gamma \phi(t)}, \; dt - a.e.$$

where $K(\cdot), \phi(\cdot)$ are given by (52) and (65), respectively, and

$$\psi(t) = \frac{1}{\gamma} e^{-\int_t^T \phi(\tau) d\tau} \int_t^T e^{\int_l^T \phi(\tau) d\tau} \left( \phi(l) \ln \left( e^{-(T-l)\delta_0} a \right) + \frac{1}{2} r(l)^\top \Sigma(l) r(l) - r_0(l) \right) dl.$$

Notice that our solutions given above coincide with the optimal solutions of classical Merton portfolio problem (see, e.g., Marín-Solano and Navas (2010) in the case with constant discount rate). This confirms the well-known fact that the time-consistent equilibrium strategy for an exponential discount function is nothing but the optimal strategy. A relevant remark is that the portfolio rule is independent of the discount factor, and it is the same for a non-exponential discount function.

### 6.2. Exponential Discounting with Non-Constant Discount Rate (Karp's Model)

Now, following Karp (2007), let us assume that the instantaneous discount rate is non-constant, but a continuous and positive function of time $\delta(l)$, for $l \in [0, T]$. Impatient agents will be characterized by a non-increasing discount rate $\delta(\cdot)$. The discount factor used to evaluate a payoff at times $\tau \geq 0$, is given by

$$\lambda(\tau) = e^{-\int_0^\tau \delta(l)dl}. \tag{69}$$

In this case, the objective is exactly the same as Marín-Solano and Navas (2010), in which the equilibrium is, however, defined within the class of feedback controls. In Marín-Solano and Navas (2010), the (feedback) equilibrium consumption-investment solutions (also called the sophisticated consumption-investment strategies) are summarized as:

(1) Logarithmic utility

$$\hat{c}(t) = \frac{1}{ae^{-\int_0^{T-t} \delta(\tau)d\tau} + \int_t^T e^{-\int_0^{l-t} \delta(\tau)d\tau}dl} \hat{X}(t), \; dt - a.e., \tag{70}$$

$$\hat{u}_I(t) = \Sigma(t)r(t)\hat{X}(t), \; dt - a.e. \tag{71}$$

(2) Power utility

$$\hat{c}(t) = (\alpha(t))^{\frac{1}{\gamma-1}}\hat{X}(t), \; dt - a.e., \tag{72}$$

$$\hat{u}_I(t) = \Sigma(t)r(t)\frac{\hat{X}(t)}{(1-\gamma)}, \; dt - a.e., \tag{73}$$

where $\alpha(\cdot)$ is the solution of the integro-differential equation,

$$\begin{cases} \alpha_t(t) - (\delta(T-t) - K(t))\alpha(t) + (1-\gamma)\alpha(t)^{\frac{\gamma}{1-\gamma}} \\ -\int_t^T e^{-\int_0^{s-t} \delta(l)dl}(\delta(s-t) - \delta(T-t))\alpha(s)^{\frac{\gamma}{1-\gamma}}e^{\gamma\int_t^s \Delta(\tau)d\tau}ds = 0, \\ \alpha(T) = a. \end{cases} \tag{74}$$

with $K(t)$ given by (52) and

$$\Delta(\tau) = r_0(\tau) + \frac{1}{(1-\gamma)}r(\tau)^\top\Sigma(\tau)r(\tau) - \alpha(\tau)^{\frac{1}{1-\gamma}}.$$

(3) Exponential utility

$$\hat{c}(t) = \phi(t)\hat{X}(t) + C(t) - \frac{\ln(\gamma a\phi(t))}{\gamma}, \; dt - a.e., \tag{75}$$

$$\hat{u}_I(t) = \Sigma(t)r(t)\frac{1}{\gamma\phi(t)}, \; dt - a.e., \tag{76}$$

where $\phi(\cdot)$ is given by (65) and $C(\cdot)$ satisfies the following very complicated integro-differential equation,

$$
\begin{cases}
C_t(t) - C(t)\phi(t) + \dfrac{1}{\gamma}\phi(t)\ln(a\gamma\phi(t)) + \dfrac{1}{2\gamma}r(t)^\top\Sigma(t)r(t) \\[2mm]
+\dfrac{1}{\gamma}\{\delta(T-t) - \phi(t) - \mathcal{K}(C(t),t)\} = 0, \\[2mm]
C(T) = 0,
\end{cases}
\tag{77}
$$

where

$$
\mathcal{K}(C(t),t) = -\mathbb{E}\Bigg[\int_t^T e^{-\int_0^{s-t}\delta(l)dl}\{\delta(s-t) - \delta(T-t)\}\phi(t)
$$
$$
\times\, e^{-\gamma\left\{C(s)-C(t)+\int_t^s \phi(\tau)Z(\tau)d\tau + \int_t^s \frac{1}{\gamma}r(\tau)^\top\Sigma(\tau)\boldsymbol{\sigma}(\tau)dW(\tau)\right\}}ds\Bigg],
\tag{78}
$$

with

$$
Z(\tau) = \frac{1}{\gamma\phi(\tau)}r(\tau)^\top\Sigma(\tau)r(\tau) - C(\tau) + \frac{1}{\gamma}\ln(\gamma a\phi(\tau)).
$$

Our (open-loop) equilibrium solutions reduce to

(1) Logarithmic utility

$$
\hat{c}(t) = \frac{1}{ae^{-\int_0^{T-t}\delta(\tau)d\tau} + \int_t^T e^{-\int_{T-l}^{T-t}\delta(\tau)d\tau}dl}\hat{X}(t),\ dt - a.e.,
\tag{79}
$$

$$
\hat{u}_I(t) = \Sigma(t)r(t)\hat{X}(t),\ dt - a.e.
\tag{80}
$$

(2) Power utility

$$
\hat{c}(t) = \frac{\left(ae^{-\int_0^{T-t}\delta(\tau)d\tau}\right)^{\frac{1}{\gamma-1}}e^{\int_t^T \frac{K(\tau)}{\gamma-1}d\tau}}{\left(1 + \int_t^T\left(ae^{-\int_0^{T-\tau}\delta(\tau)d\tau}\right)^{\frac{1}{\gamma-1}}e^{\int_\tau^T \frac{K(l)}{\gamma-1}dl}d\tau\right)}\hat{X}(t),\ dt - a.e.,
\tag{81}
$$

$$
\hat{u}_I(t) = \Sigma(t)r(t)\frac{\hat{X}(t)}{(1-\gamma)},\ dt - a.e.
\tag{82}
$$

(3) Exponential utility

$$
\hat{c}(t) = -\frac{1}{\gamma}\ln\left(ae^{-\int_0^{T-t}\delta(\tau)d\tau}\right) + \phi(t)\hat{X}(t) + \psi(t),\ dt - a.e.,
\tag{83}
$$

$$
\hat{u}_I(t) = \Sigma(t)r(t)\frac{1}{\gamma\hat{X}(t)\phi(t)},\ dt - a.e.,
\tag{84}
$$

where $K(\cdot)$, $\phi(\cdot)$ are given by (52) and (65), respectively, and

$$
\psi(t) = e^{-\int_t^T\phi(\tau)d\tau}\int_t^T e^{\int_l^T\phi(\tau)d\tau}\left(\frac{1}{\gamma}\phi(l)\ln\left(e^{-\int_0^{T-t}\delta(\tau)d\tau}a\right) + \frac{1}{2\gamma}r(l)^\top\Sigma(l)r(l) - \frac{r_0(l)}{\gamma}\right)dl.
$$

**Remark 5.** *Comparing the results of this special case with our solutions, we find the following facts: The equilibrium proportion investment strategies coincide in the three cases. The consumption strategies are different in the three cases. Moreover, our equilibrium consumption strategies are well defined and explicitly given, while, in Marín-Solano and Navas (2010), equilibrium consumption strategies in the case of Power utility, as well as in the case of Exponential utility, are obtained via a very complicated integro-differential equations, in which unique solvability are not established.*

**Author Contributions:** All authors have contributed equally. All authors have read and agreed to the published version of the manuscript.

**Funding:** This research received no external funding.

**Acknowledgments:** The authors are particularly grateful for the editor and anonymous reviewers for their valuable and constructive suggestions.

**Conflicts of Interest:** The authors declare no conflict of interest.

## Appendix A

We derive the proof of Proposition 1 by means of the duality analysis and some limiting procedures. Moreover, since our objective function is not in quadratic form, we need to adapt some results obtained in Hu et al. (2012) and Hu et al. (2015) according to our control problem which concerns a general and non-necessary quadratic utility maximization.

**Proof of Proposition 1.** The estimates (25)–(27) follow from Theorem 4.4 in Yong and Zhou (1999). Moreover, the following representation holds for the objective functional:

$$
\begin{aligned}
&J\big(t, \hat{X}(t), u^\varepsilon(\cdot)\big) - J\big(t, \hat{X}(t), \hat{u}(\cdot)\big) \\
&= \mathbb{E}^t\left[\int_t^T \lambda(s-t)\big(\varphi(\Gamma^\top u^\varepsilon(s)) - \varphi(\Gamma^\top \hat{u}(s))\big)ds + \lambda(T-t)\big(h(\hat{X}^\varepsilon(T)) - h(\hat{X}(T))\big)\right].
\end{aligned}
\tag{A1}
$$

Now, from (22) and by applying the second order Taylor-Young expansion, we find that

$$
\begin{aligned}
h\big(\hat{X}^\varepsilon(T)\big) - h\big(\hat{X}(T)\big) &= h_x\big(\hat{X}(T)\big)(y^{\varepsilon,v}(T) + z^{\varepsilon,v}(T)) + \frac{1}{2}h_{xx}\big(\hat{X}(T)\big)(y^{\varepsilon,v}(T) + z^{\varepsilon,v}(T))^2 \\
&\quad + o\Big((y^{\varepsilon,v}(T) + z^{\varepsilon,v}(T))^2\Big).
\end{aligned}
$$

By applying the second order Taylor-Lagrange expansion, we get

$$
\varphi\big(\Gamma^\top u^\varepsilon(s)\big) - \varphi\big(\Gamma^\top \hat{u}(s)\big) = \Big\langle \varphi_x\big(\Gamma^\top \hat{u}(s)\big)\Gamma, v\Big\rangle + \frac{1}{2}\Big\langle \varphi_{xx}\big(\Gamma^\top \hat{u}(s) + \theta v 1_{[t,t+\varepsilon]}\big)\Gamma\Gamma^\top v, v\Big\rangle.
$$

From (27), it holds that

$$
\begin{aligned}
&J\big(t, \hat{X}(t), u^\varepsilon(\cdot)\big) - J\big(t, \hat{X}(t), \hat{u}(\cdot)\big) \\
&= \mathbb{E}^t\Bigg[\int_t^T \lambda(s-t)\Big\{\big\langle \varphi_x(\Gamma^\top \hat{u}(s))\Gamma, v\big\rangle + \tfrac{1}{2}\big\langle \varphi_{xx}\big(\Gamma^\top \hat{u}(s) + \theta v 1_{[t,t+\varepsilon]}\big)\Gamma\Gamma^\top v, v\big\rangle\Big\}1_{[t,t+\varepsilon]}ds \\
&\quad + \lambda(T-t)\Big(h_x\big(\hat{X}(T)\big)(y^{\varepsilon,v}(T) + z^{\varepsilon,v}(T)) + \tfrac{1}{2}h_{xx}\big(\hat{X}(T)\big)(y^{\varepsilon,v}(T) + z^{\varepsilon,v}(T))^2\Big)\Bigg] \\
&\quad + o(\varepsilon).
\end{aligned}
\tag{A2}
$$

Notice that

$$
\begin{aligned}
&\lambda(T-t)\left(h_x\big(\hat{X}(T)\big)(y^{\varepsilon,v}(T) + z^{\varepsilon,v}(T)) + \frac{1}{2}h_{xx}\big(\hat{X}(T)\big)(y^{\varepsilon,v}(T) + z^{\varepsilon,v}(T))^2\right) \\
&= p(T;t)(y^{\varepsilon,v}(T) + z^{\varepsilon,v}(T)) + \frac{1}{2}P(T;t)(y^{\varepsilon,v}(T) + z^{\varepsilon,v}(T))^2.
\end{aligned}
$$

Now, by applying Itô's formula to $s \mapsto p(s;t)(y^{\varepsilon,v}(s) + z^{\varepsilon,v}(s))$ on $[t, T]$, we get

$$
\mathbb{E}^t[p(T;t)(y^{\varepsilon,v}(T) + z^{\varepsilon,v}(T))] = \mathbb{E}^t\left[\int_t^{t+\varepsilon}\big\{v^\top B(s)p(s;t) + v^\top D(s)\widetilde{q}(s;t)\big\}ds\right].
\tag{A3}
$$

Again, by applying Itô's formula to $s \mapsto P(s;t)(y^{\varepsilon,v}(s) + z^{\varepsilon,v}(s))^2$ on $[t,T]$, we get

$$
\begin{aligned}
\mathbb{E}^t & \left[ P(T;t)(y^{\varepsilon,v}(T) + z^{\varepsilon,v}(T))^2 \right] \\
& = \mathbb{E}^t \left[ \int_t^{t+\varepsilon} \left\{ 2v^\top (y^{\varepsilon,v}(s) + z^{\varepsilon,v}(s)) \left( B(s)P(s,t) + D(s)\widetilde{Q}(s,t) \right) \right. \right. \\
& \qquad \left. \left. + v^\top \left( D(s)D(s)^\top \right) v P(s,t) \right\} ds \right],
\end{aligned}
\tag{A4}
$$

where $\widetilde{Q}(s;t) = \left( 0, Q(s;t)^\top \right)^\top$. On the other hand, we conclude from **(H1)**, together with (27), that

$$
\mathbb{E}^t \left[ \int_t^{t+\varepsilon} (y^{\varepsilon,v}(s) + z^{\varepsilon,v}(s)) \left( B(s)P(s,t) + D(s)\widetilde{Q}(s,t) \right) ds \right] = o(\varepsilon).
\tag{A5}
$$

By taking $(A.1.4)$, $(A.1.5)$, and $(A.1.6)$ in $(A.1.3)$, it follows that

$$
\begin{aligned}
& J\left(t, \hat{X}(t), u^\varepsilon(\cdot)\right) - J\left(t, \hat{X}(t), \hat{u}(\cdot)\right) \\
& = \mathbb{E}^t \left[ \int_t^{t+\varepsilon} \left\{ \left\langle B(s)p(s;t) + D(s)\widetilde{q}(s;t) + \lambda(s-t)\varphi_x\left(\Gamma^\top \hat{u}(s)\right)\mathbf{1}_{[t,t+\varepsilon)}\Gamma, v \right\rangle \right. \right. \\
& \qquad \left. \left. + \tfrac{1}{2} \left\langle \left( \lambda(s-t)\varphi_{xx}\left( \left\langle \Gamma, \hat{u}(s) + \theta v 1_{[t,t+\varepsilon)} \right\rangle \right) \right) \Gamma\Gamma^\top + P(s,t)D(s)D(s)^\top \right) v, v \right\rangle \right\} ds \right] + o(\varepsilon),
\end{aligned}
$$

which is equivalent to (28).

Now, we derive the proof of Lemma 1 by using some limiting procedures. First, let us recall the following lemma which was proved by Wang (2020) (Lemma 3.3).  □

**Lemma A1.** *If $\phi(\cdot) = (\phi_1(\cdot), \dots, \phi_m(\cdot)) \in \mathcal{M}_{\mathcal{F}}^p(0,T;\mathbb{R}^m)$ with $m \in \mathbb{N}$ and $p > 1$, then, for a.e. $t \in [0,T)$, there exists a sequence $\{\varepsilon_n^t\}_{n\in\mathbb{N}} \subset (0, T-t)$ depending on $t$ such that $\lim\limits_{n\to\infty} \varepsilon_n^t = 0$ and*

$$
\lim_{n\to\infty} \frac{1}{\varepsilon_n^t} \mathbb{E}^t \left[ \int_t^{t+\varepsilon_n^t} |\phi_i(s) - \phi_i(t)|^p ds \right] = 0, \text{ for } i = 1, \dots, m, \, d\mathbb{P} - a.s.
$$

**Proof of Lemma 1.** We define, for $t \in [0,T]$ and $s \in [t,T]$,

$$
(\bar{p}(s;t), \bar{q}(s;t)) := \frac{1}{\lambda(T-t)} e^{-\int_s^T r_0(\tau)d\tau} (p(s;t), q(s;t)).
$$

Then, for any $t \in [0,T]$, in the interval $[t,T]$, the pair $(\bar{p}(\cdot;t), \bar{q}(\cdot;t))$ satisfies

$$
\begin{cases}
d\bar{p}(s;t) = \bar{q}(s;t)^\top dW(s), \ s \in [t,T], \\
\bar{p}(T;t) = h_x(\hat{X}(T)).
\end{cases}
\tag{A6}
$$

Moreover, it is clear that, from the uniqueness of solutions to $(A.2.1)$, we have the equality $(\bar{p}(s;t_1), \bar{q}(s;t_1)) = (\bar{p}(s;t_2), \bar{q}(s;t_2))$, for any $t_1, t_2, s \in [0,T]$ such that $0 < t_1 < t_2 < s < T$. Hence, the solution $(\bar{p}(\cdot;t), \bar{q}(\cdot;t))$ does not depend on the variable $t$, and this allows us to denote the solution of $(A.2.1)$ by $(\bar{p}(\cdot), \bar{q}(\cdot))$.

We have then, for any $t \in [0,T]$, and $s \in [t,T]$,

$$
(p(s;t), q(s;t)) = \lambda(T-t)e^{\int_s^T r_0(\tau)d\tau} (\bar{p}(s), \bar{q}(s)).
\tag{A7}
$$

Now, using $(A.2.2)$ we have, under **(H2)**, for any $t \in [0,T]$ and $s \in [t,T]$,

$$
|p(s;t) - p(s;s)| \leq \sup_{t \leq s \leq t+\varepsilon} |\lambda(T-t) - \lambda(T-s)| e^{-\int_s^T r_0(\tau)d\tau} |\bar{p}(s)|,
\tag{A8}
$$

and

$$|q(s;t) - q(s;s)| \leq \sup_{t \leq s \leq t+\varepsilon} |\lambda(T-t) - \lambda(T-s)| e^{-\int_s^T r_0(\tau)d\tau} |\bar{q}(s)|. \tag{A9}$$

From which, we have, for any $a > 0$, $t \in [0, T]$, and $\varepsilon \in (0, T - t)$,

$$
\begin{aligned}
&\mathbb{P}\left( \left| \frac{1}{\varepsilon} \mathbb{E}^t \left[ \int_t^{t+\varepsilon} \mathcal{H}(s;t)ds \right] - \frac{1}{\varepsilon} \mathbb{E}^t \left[ \int_t^{t+\varepsilon} \mathcal{H}(s;s)ds \right] \right| \geq a \right), \\
&\leq \frac{1}{a} \mathbb{E} \left| \frac{1}{\varepsilon} \mathbb{E}^t \left[ \int_t^{t+\varepsilon} \mathcal{H}(s;t)ds \right] - \frac{1}{\varepsilon} \mathbb{E}^t \left[ \int_t^{t+\varepsilon} \mathcal{H}(s;s)ds \right] \right|, \\
&\leq C \sup_{t \leq s \leq t+\varepsilon} |\lambda(T-t) - \lambda(T-s)| \frac{1}{\varepsilon} \mathbb{E} \int_t^{t+\varepsilon} (|\bar{p}(s)| + |\bar{q}(s)|)ds \\
&\quad + \sup_{t \leq s \leq t+\varepsilon} |\lambda(s-t) - 1| \frac{1}{\varepsilon} \int_t^{t+\varepsilon} \mathbb{E}\left[ \varphi_x\left( \Gamma^\top \hat{u}(s) \right) \right] ds.
\end{aligned}
$$

Noting that, since $\lambda(\cdot)$ is continuous, we get

$$\lim_{\varepsilon \downarrow 0} \sup_{t \leq s \leq t+\varepsilon} |\lambda(T-t) - \lambda(T-s)| = 0$$

for $t \in [0, T]$. Moreover, since $(\bar{p}(\cdot), \bar{q}(\cdot)) \in \mathcal{L}_{\mathcal{F}}^2(0, T; \mathbb{R}) \times \mathcal{M}_{\mathcal{F}}^2\left(0, T; \mathbb{R}^d\right)$, we get

$$\lim_{\varepsilon \downarrow 0} \sup_{t \leq s \leq t+\varepsilon} |\lambda(T-t) - \lambda(T-s)| \frac{1}{\varepsilon} \mathbb{E} \int_t^{t+\varepsilon} (|\bar{p}(s)| + |\bar{q}(s)|)ds = 0.$$

Noting that $\lambda(0) = 1$, then $\lim_{\varepsilon \downarrow 0} \sup_{t \leq s \leq t+\varepsilon} |\lambda(s-t) - 1| = 0$. According to **(H3)**, by using the dominated convergence theorem,

$$\lim_{\varepsilon \downarrow 0} \frac{1}{\varepsilon} \int_t^{t+\varepsilon} \mathbb{E}\left[ \varphi_x\left( \Gamma^\top \hat{u}(s) \right) \right] ds = \mathbb{E}\left[ \varphi_x\left( \Gamma^\top \hat{u}(t) \right) \right] < \infty, \, dt - a.e.$$

Therefore,

$$\lim_{\varepsilon \downarrow 0} \mathbb{E} \left| \frac{1}{\varepsilon} \mathbb{E}^t \left[ \int_t^{t+\varepsilon} \mathcal{H}(s;t)ds \right] - \frac{1}{\varepsilon} \mathbb{E}^t \left[ \int_t^{t+\varepsilon} \mathcal{H}(s;s)ds \right] \right| = 0.$$

Hence, for each $t$, there exists a sequence $(\varepsilon_n^t)_{n \geq 0} \subset (0, T-t)$ such that $\lim_{n \to \infty} \varepsilon_n^t = 0$ and

$$\lim_{n \to \infty} \left| \frac{1}{\varepsilon_n^t} \mathbb{E}^t \left[ \int_t^{t+\varepsilon_n^t} \mathcal{H}(s;t)ds \right] - \frac{1}{\varepsilon_n^t} \mathbb{E}^t \left[ \int_t^{t+\varepsilon_n^t} \mathcal{H}(s;s)ds \right] \right| = 0, \, d\mathbb{P} - a.s.$$

Moreover, since $\varphi_x\left( \Gamma^\top \hat{u}(\cdot) \right) \in \mathcal{M}_{\mathcal{F}}^p(0, T; \mathbb{R})$ and

$$(\bar{p}(\cdot), \bar{q}(\cdot)) \in \mathcal{L}_{\mathcal{F}}^2(0, T; \mathbb{R}) \times \mathcal{M}_{\mathcal{F}}^2\left(0, T; \mathbb{R}^d\right),$$

we get, from Lemma A1, that there exists a sub-sequence of $(\varepsilon_n^t)_{n \geq 0}$, which is also denoted by $(\varepsilon_n^t)_{n \geq 0}$ such that

$$\lim_{n \to \infty} \frac{1}{\varepsilon_n^t} \mathbb{E}^t \left[ \int_t^{t+\varepsilon_n^t} \mathcal{H}(s;s)ds \right] = \mathcal{H}(t;t), \, dt - a.e, \, d\mathbb{P} - a.s.$$

To derive the statement 2) in the Lemma 1, it is sufficient to prove the following, for each $t$ there exists a sequence $\left(\varepsilon_n^t\right)_{n \geq 0} \subset (0, T - t)$ such that $\lim\limits_{n \to \infty} \varepsilon_n^t = 0$ and

$$\lim_{n \to \infty} \frac{1}{\varepsilon_n^t} \mathbb{E}^t \left[ \int_t^{t+\varepsilon_n^t} \lambda(s-t) \varphi_{xx}\left( \Gamma^\top \left( \hat{u}(s) + \theta v 1_{[t,t+\varepsilon]} \right) \right) ds \right] = \varphi_{xx}\left( \Gamma^\top(\hat{u}(t)) \right),$$

$$\lim_{n \to \infty} \frac{1}{\varepsilon_n^t} \mathbb{E}^t \left[ \int_t^{t+\varepsilon_n^t} \lambda(s-t) \sigma(s)\sigma(s)^\top P(s;t) ds \right] = \sigma(t)\sigma(t)^\top P(t;t).$$

Let us prove the first limit. We have

$$\left| \frac{1}{\varepsilon} \mathbb{E}^t \left[ \int_t^{t+\varepsilon} \lambda(s-t) \varphi_{xx}\left( \Gamma^\top \left( \hat{u}(s) + \theta v 1_{[t,t+\varepsilon]} \right) \right) ds \right] - \frac{1}{\varepsilon} \mathbb{E}^t \left[ \int_t^{t+\varepsilon} \varphi_{xx}\left( \Gamma^\top(\hat{u}(s)) \right) ds \right] \right|$$

$$\leq \sup_{t \leq s \leq t+\varepsilon} |\lambda(s-t) - 1| \frac{1}{\varepsilon} \mathbb{E}^t \left[ \int_t^{t+\varepsilon} \sup_{\eta \leq M} \left| \varphi_{xx}\left( \Gamma^\top(\hat{u}(s) + \eta) \right) \right| ds \right].$$

Applying the same arguments used in the first limit, we obtain, according to Lemma A1,

$$\lim_{n \to \infty} \frac{1}{\varepsilon_n^t} \mathbb{E}^t \left[ \int_t^{t+\varepsilon_n^t} \varphi_{xx}\left( \Gamma^\top(\hat{u}(s)) \right) ds \right] = \varphi_{xx}\left( \Gamma^\top(\hat{u}(t)) \right),$$

at least for a sub-sequence. $\square$

## Appendix B. Conclusions

It is well known that the inconsistent optimal control problems are complicated to solve in general. In this paper, we have studied optimal investment and consumption problem, since the objective functional may depend on the non-exponential discount function. So, inspired from the previous literature, we made some restrictive conditions on the coefficients of the utility function, which allows to treat some limiting procedures in the original problem.

We have used the game theoretic approach to handle the time inconsistency. Specifically, feedback Nash equilibrium controls are explicitly constructed as an alternative of optimal controls. This has been accomplished through the necessary and sufficient condition for equilibrium during stochastic system that includes a flow of forward-backward stochastic differential equations. We derive the closed-from expression of the equilibrium investment-consumption strategy. Moreover, some particular cases of our model are discussed and compared with the previous literature from the extended HJB equations with a verification theorem.

The work can be extended in several ways. For example, this approach can be extended to a general continuous-time stochastic control problem with delay under a fairly general time-inconsistent objective functional. Another challenging problem, is the study of some statistical testing method for validation of the smoothness assumptions about the coefficients; see, e.g., Pešta and Wendler (2020). The research on these topics is in progress and will appear in our forthcoming papers.

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
