# Peer review of "Time-Consistent Investment and Consumption Strategies under a General Discount Function"

_jrfm, doi:10.3390/jrfm14020086_

Round 1

Reviewer 1 Report

This study investigates equilibrium solutions for a time-inconsistent consumption-investment problem with a non-exponential discount function and a general utility function. I would like to thank the authors for bringing this issue. This is through manuscript: well written and presented. However, the conclusion and limitations section is missing in the paper. More importantly, I would love to see empirical evidence utilizing this approach to see how the proposed approach works in the real data world. Great jobs. 

Author Response

The reply is enclosed.

Reviewer 2 Report

The Merton portfolio management problem in the context of non-exponential discounting is investigated in this manuscript. Such a context causes time-inconsistency of the decision maker. Equilibrium policies within the class of open-loop controls are considered. Special cases of power, logarithmic and exponential utility functions for the explicit representation of the equilibrium policies are discussed.

I consider this paper a nice contribution to the investment-consumption problems under discounted utility. Therefore, the manuscript is worth to be published.

My major remark is that Assumptions (H1)-(H4) should be discussed in more detail from a practical as well as from a theoretical point of view. For instance, a statistical testing method for possible change points can be utilized (and should be referred) for validation of the smoothness assumptions, cf. Pešta, M. and Wendler, M. (2020). Nuisance-parameter-free changepoint detection in non-stationary series. TEST, 29(2):379-408.

Author Response

The reply is enclosed

Reviewer 3 Report

The paper is prepared with perfect methodological background, representing a very high quality. The topic is absolutely relevant for the journal.

I recommend improvements as follows:

  • The intruduction chapter should be restructured, separating the literature review in a separate chapter.
  • This literature review chapter should be longer, including and critically analysing the sources of famous scientists in the field.
  • A short methodology chapter should be formulated, highligting the methods and materials used. It is just an issue of restructuring, too.
  • At the end, conclusions, recommendations and limitations shoud be described, these are missing now.

Author Response

The reply is enclosed
